# Imaging mitochondrial membrane potential via concentration-dependent fluorescence lifetime changes

Dilizhatai Saimi[1], Luc Reymond [2], Tursunjan Aziz[3], Xuan Shen[4], Ziying Luo[1,5], Shuaibo Pi[1], Yitong Liu[1,5], Song Fu[6,7,8], Shuangjin Ding[1], Anming Meng [3,9,10], Liangyi Chen [1,11,12], Hui Jiang [7,8,13] & Zhixing Chen [1,5,11,12,14] ✉

Mitochondria are central to cellular metabolism. Various fluorescence tools have been developed for imaging the mitochondrial environment. Yet, new reporters and imaging methods for directly reading the mitochondrial status are needed for high spatial-temporal resolution imaging. Here, we introduce PK Mito Deep Red (PKMDR), a low-phototoxicity mitochondrial probe for time-lapse imaging, whose fluorescence lifetime serves as a sensitive indicator of mitochondrial membrane potential ($\Delta\psi_m$). The positively charged PKMDR accumulates within mitochondria under a higher $\Delta\psi_m$, leading to concentration-induced quenching and a measurable decrease in fluorescence lifetime. Since mitochondrial respiration primarily regulates $\Delta\psi_m$, PKMDR's fluorescence lifetime effectively reports on the status of oxidative phosphorylation. Using PKMDR with fluorescence lifetime imaging microscopy (FLIM), we visualize heterogeneous $\Delta\psi_m$ across individual cells, organoids, and tissues over time. This method reliably reveals the heterogeneity between metabolically active peripheral mitochondria and relatively inactive perinuclear mitochondria in various cell types. Overall, PKMDR-FLIM is a robust tool for directly visualizing $\Delta\psi_m$ with high spatiotemporal resolution.

Mitochondria are essential organelles that serve as powerhouses of the cell and act as transition hubs for multiple cellular signaling pathways, such as autophagy and apoptosis. Mitochondrial dynamics are closely correlated with cellular activities. Thus, regulating their plasticity and dynamic features is crucial for controlling cellular respiration levels and maintaining homeostasis in organisms[1,2]. In oxidative phosphorylation (OXPHOS), ATP synthase and the electron transport chain (ETC) generate ATP, which relies on mitochondrial membrane potential (MMP, or $\Delta\psi_m$) for the proton-motive force. High-energy electron carriers (NADH and FADH$_2$) produced in the Krebs cycle sequentially

[1]College of Future Technology, Institute of Molecular Medicine, National Biomedical Imaging Center, Beijing Key Laboratory of Cardiometabolic Molecular Medicine, Peking University, Beijing, China. [2]Biomolecular Screening Facility, École Polytechnique Fédérale de Lausanne (EPFL), Lausanne, Switzerland. [3]Laboratory of Molecular Developmental Biology, State Key Laboratory of Membrane Biology, Tsinghua-Peking Center for Life Sciences, School of Life Sciences, Tsinghua University, Beijing, China. [4]College of Chemistry and Molecular Engineering, Synthetic and Functional Biomolecules Center, Beijing National Laboratory for Molecular Sciences, Key Laboratory of Bioorganic Chemistry and Molecular Engineering of Ministry of Education, Peking University, Beijing, China. [5]Peking-Tsinghua Center for Life Science, Academy for Advanced Interdisciplinary Studies, Peking University, Beijing, China. [6]Graduate School of Peking Union Medical College, Beijing, China. [7]National Institute of Biological Sciences, Beijing, China. [8]Beijing Key Laboratory of Cell Biology for Animal Aging, Beijin, China. [9]Developmental Diseases and Cancer Research Center, Sun Yat-sen Memorial Hospital, Sun Yat-sen University, Guangzhou, China. [10]Laboratory of Stem Cell Regulation, Guangzhou Laboratory, Guangzhou, China. [11]PKU-Nanjing Institute of Translational Medicine, Nanjing, China. [12]State Key Laboratory of Membrane Biology, Peking University, Beijing, China. [13]Tsinghua Institute of Multidisciplinary Biomedical Research, Tsinghua University, Beijing, China. [14]GenVivo Tech, Nanjing, China. ✉e-mail: zhixingchen@pku.edu.cn

transfer their electrons to oxygen along the chain through ETC complexes, while protons are pumped out of the matrix by harnessing the released energy, establishing MMP between the intermembrane space and the matrix[3].

Alterations in $\Delta\psi_m$ can disrupt proton gradient formation, leading to metabolic disturbances[4]. Therefore, monitoring the membrane potential across the inner membrane is crucial for assessing the metabolic activity of mitochondria and cell viability. Classical MMP probes are cationic small-molecule dyes, such as Rhodamine 123[5], TMRM (tetramethyl rhodamine methyl ester), TMRE[6], and JC-1[7], whose accumulation in the inner membrane is proportional to $\Delta\psi_m$ according to the Nernst equation. Among these, TMRM and TMRE are widely regarded as "gold standards" for measuring $\Delta\psi_m$ through fluorescence intensity readouts[8,9]. However, intensity readouts may produce misleading information due to inner filter effects—absorption and scattering events that modulate detected fluorescence intensity[10]. Additionally, their application is susceptible to concentration fluctuations, sample geometry, and photobleaching[10]. The J-aggregate-induced spectral shifts of the JC-1 molecule render it a ratiometric probe capable of reading absolute potential via an intensive parameter rather than an extensive one. However, the JC-1 protocol suffers from weak signals and cumbersome calibrations[8].

The fluorescence lifetime (τ) of a fluorophore is the average time it spends in the excited state before returning to the ground state. For a fluorescent probe, τ serves as a kinetic parameter that indicates the intensity decreases to 1/e of its initial value. Fluorescence lifetime imaging microscopy (FLIM) is gaining increasing attention in biological imaging. Tailored FLIM sensors have been developed for various parameters, such as pH[11], viscosity[12,13], temperature[14] and calcium[15]. Compared to intensity imaging, FLIM has the advantage of measuring absolute values, as lifetime is an intensive parameter whose value is not affected by fluorescence signal strength[16]. Therefore, FLIM can bypass the drawback of intensity imaging to a certain extent[17].

Recently, scientists have developed several strategies to study mitochondria using FLIM. For instance, Mito Flipper has been reported to respond to osmotic shocks and provide membrane tension readouts[18]. MitoPB Red indicates polarity changes under oxidative stress[19]. Classic dyes such as TMRM and SYTO are also responsive under FLIM, with their signals attributed to mitochondrial membrane potential[20]. Moreover, the fluorescence lifetime of Mitorotor-1 is affected by molecular free volume, which relates to lateral diffusion and is further subject to change in IMM fluidity[21]. Nonetheless, these probes have not yet established a clear-cut relationship between the metabolic state of mitochondria and the fluorescence lifetime of probes, and their versatility with advanced samples has not been fully demonstrated.

Here, we leverage PKMDR, a gentle far-red mitochondrial probe developed by our group, to precisely report mitochondrial metabolic heterogeneity from individual cells to tissues using fluorescence lifetime imaging technology. PKMDR offers high brightness and very low phototoxicity, making it ideal for long-term time-lapse recordings[22]. Although it is commonly accepted that fluorescence lifetime is independent of the concentration of fluorescent molecules, we demonstrate that the fluorescence lifetime of PKMDR decreases at high concentrations due to concentration-induced quenching. Since the mitochondrial accumulation of PKMDR depends on mitochondrial membrane potential, the fluorescence lifetime of PKMDR can directly reflect the membrane potential of mitochondria. We showcase advanced applications under different metabolic conditions, including senescence, knockout cells, oocytes at various stages, and T cells before and after activation. Additionally, PKMDR reveals metabolic heterogeneity within individual cells, across different cell cycles, between individual cells of embryos, and in organoid and tumor tissues.

## Result

### Cationic dyes generally exhibit concentration-dependent fluorescence lifetime due to quenching

In our earlier study, we devised PK Mito dyes, which were red and far-red probes specifically designed for mitochondrial imaging[22,23]. Thanks to the alleviated phototoxicity rendered by the intramolecular triplet state quenchers, PK Mito dyes demonstrated minimal levels of phototoxicity, which was ideal for time-lapse imaging. Among them, PK Mito Deep Red (PKMDR) was based on the far-red chromophore Cy5, which was particularly prone to aggregation[24,25]. A previous study has shown that Cy5 on a heavily labeled secondary antibody, with a degree of labeling (DOL) greater than 3, exhibited a lower fluorescence lifetime compared to antibodies with a DOL of 1[26]. This prior data led us to speculate that PKMDR may display a concentration-dependent fluorescence lifetime (Fig. 1a), especially in mitochondrial imaging.

To test this hypothesis, we measured the fluorescence intensity and lifetime of PKMDR at various concentrations of DMSO using a fluorescence spectrophotometer, whose cuvette-based setup offers accurate measurements. The relationship between fluorescence intensity and dye concentration deviated from linearity starting at 20 μM, indicating self-quenching behavior (Supplementary Fig. 1a)[27]. Accordingly, the fluorescence lifetime of PKMDR sharply declined above 20 μM (Fig. 1b), collectively supporting the emergence of concentration-induced quenching. Notably, while PKMDR was mainly in a dispersed state in solution below 20 μM, there is a known artifact of radiative energy transport (RET). Within the long light path of the cuvette, the re-absorption of the emitted fluorescence photon at another dye molecule would give rise to an apparent increase in lifetime, accounting for the fluorescence lifetime appeared to increase between 1 μM and 20 μM[17,28]. At concentrations above 20 μM, as the PKMDR molecules started to self-quench, the fluorescence lifetime decreased with increasing concentration, yet the total fluorescence intensity was still going upwards (Fig. 1c). At above 500 μM, the fluorescence intensity of the solution finally decreased due to a strong quenching effect[17].

The counterintuitive variation in fluorescence lifetimes with concentration revealed alterations in the photophysical properties of the fluorophore at elevated concentrations, potentially indicative of behavior arising during the aggregation process of this fluorescent molecule. We speculated that such behavior should be general among fluorescent dyes[29]. Concentration-induced quenching can also be recorded with the DMSO solution of TMRM and TMRE, which gave a decreasing fluorescence lifetime at mM concentration ranges (Fig. 1d, Supplementary Fig. 1b). Notably, TMRM and TMRE were less hydrophobic than PKMDR, which translates into higher critical concentrations of fluorescence lifetime drop.

We further established the generality of concentration-caused lifetime changes in membranous environment, where the fluorescence lifetime of PKMDR in giant unilamellar vesicles (GUVs) at various concentrations was measured using FLIM imaging (Fig. 1e). GUVs were prepared using the electro-formation technique, where dyes were doped into the GUVs solution. Of note, the concentration of dyes referred to the apparent concentrations in the GUVs solution, as the absolute concentration of dyes in the 2D lipid bilayer was hard to quantify. In GUVs, PKMDR lifetime showed a consistent decrease as the apparent concentration of the dye increased (Fig. 1g). Mitorotor-1, whose fluorescence lifetime was suggested as a microviscosity sensor[21], gave a concentration-dependent lifetime in GUVs as well (Fig. 1f, h). Such a trend was also generalizable to TMRM and TMRE in the GUVs (Fig. 1i, j and Supplementary Fig. 2), although occurring at significantly higher dye concentrations. Notably, PKMDR exhibited the largest dynamic range of fluorescence lifetime variation among the tested dyes, spanning from 0.2 ns to 1.5 ns across the concentration range tested (Fig. 1g–j). Moreover, we also examined the impact of temperature and pH on the lifetime of PKMDR. The fluorescence lifetime of PKMDR was measured in phosphate buffer solutions across pH

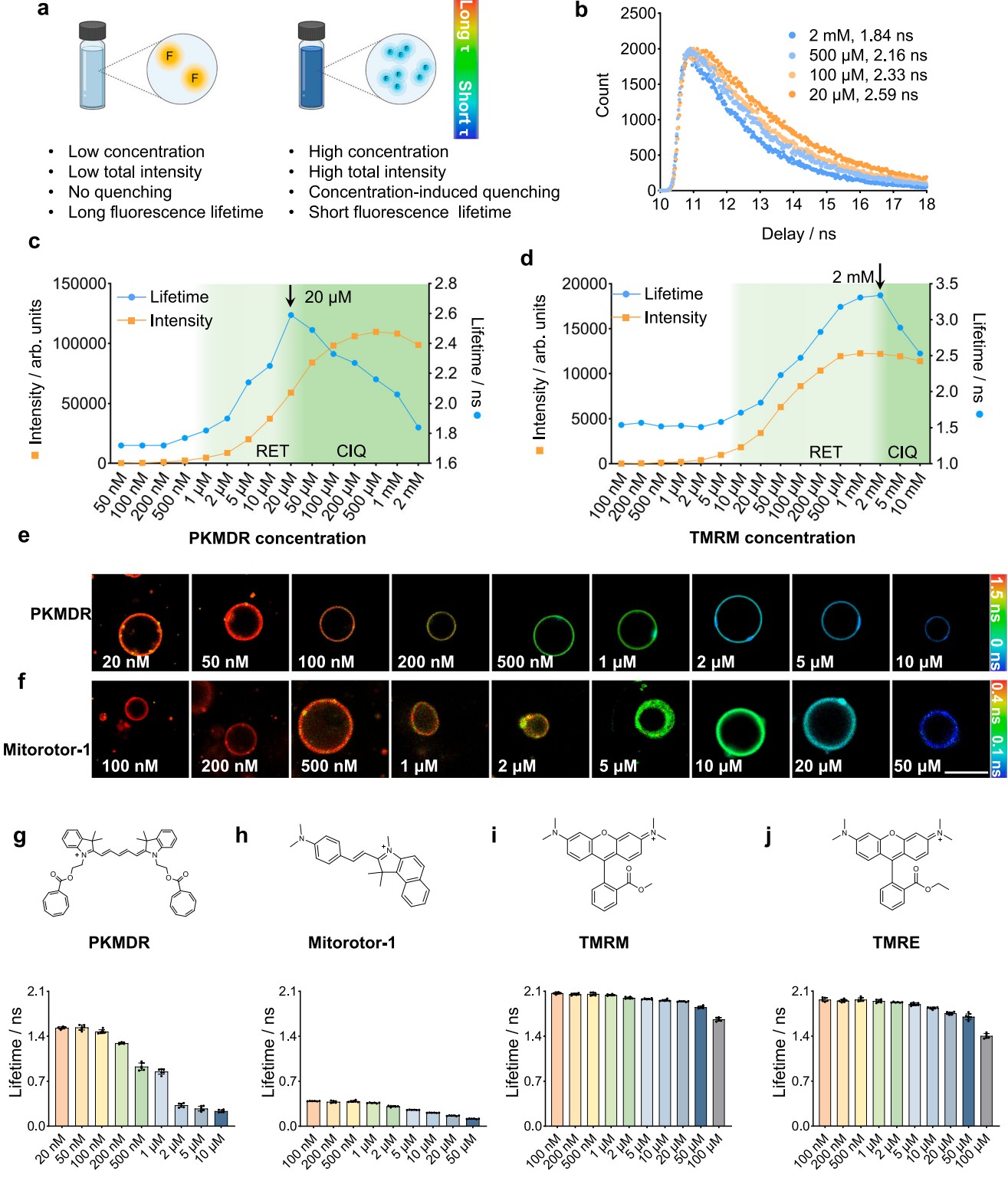

**Fig. 1 | Cationic dyes at high concentrations decreased the apparent fluorescence lifetime. a** Schematic illustration of fluorescent dyes exhibits concentration-dependent fluorescence lifetime. **b** Fluorescence lifetime decay curves of PKMDR in DMSO solution at different concentrations. **c, d** Fluorescence intensity (orange dot) and lifetime (blue dot) of PKMDR (**c**) and TMRM (**d**) in DMSO at various concentrations showing a fluorescence lifetime decrease. **e, f** FLIM images of 1, 2 dioleoyl sn-glycero-3-phosphocholine (DOPC) GUVs stained with PKMDR (**e**) and Mitorotor-1 (**f**) at various dye concentrations. Scale bars, 10 μm. **g–j** Fluorescence lifetime plot of PKMDR (**g**), Mitorotor-1 (**h**), TMRM (**i**), TMRE (**j**) doped in DOPC giant unilamellar vesicles (GUVs) at various concentrations. Data were presented as the mean ± SEM, $n = 6$ liposomes. RET radiative energy transport, CIQ concentration-induced quenching.

5.8–7.8. The results demonstrated that the fluorescence lifetime remained stable and independent of environmental pH within this physiologically relevant range (Supplementary Fig. 3a). To evaluate temperature effects, PKMDR was incorporated into DOPC vesicles. The fluorescence lifetime exhibited a temperature-dependent change of 0.38 ns over the range of 25 °C to 45 °C (Supplementary Fig. 3b). Importantly, this variation is significantly smaller than the 1.24 ns lifetime shift induced by concentration-dependent aggregation of PKMDR (Fig. 1g). While temperature modulates the fluorescence lifetime to a measurable extent, its contribution is ~3.3-fold weaker compared to concentration-dependent effects. This indicates that PKMDR's sensing performance is primarily governed by its concentration rather than thermal fluctuations under physiological conditions. Overall, these data suggested that the concentration-dependent fluorescence lifetime was a general phenomenon in cationic mitochondrial dyes, where PKMDR was a potential FLIM sensor due to its low critical concentration, high brightness, and large dynamic range in fluorescence lifetime (Supplementary Table 1).

## PKMDR is a FLIM probe that reveals the mitochondrial inner membrane potential

We then exploited the concentration-based lifetime change of PKMDR in mitochondrial imaging. PKMDR was an established inner-membrane stain, as confirmed with two-color SIM imaging of U2OS cells using Tomm20-GFP as an outer-membrane counter-marker (Fig. 2a). PKMDR, like many cationic dyes, exhibits a high affinity to mitochondria due to the electrochemical drive of the negative potential of mitochondrial membrane potential (MMP). In contrast to TMRM mitochondrial dye, which mainly accumulates in the mitochondrial matrix, the hydrophobic nature of PKMDR renders its selective partition at the inner mitochondria membrane (IMM), giving a higher concentration and an amplified change-of-concentration in the 2D compartment. Therefore, we speculate that the concentration-induced lifetime change of PKMDR can be a sensitive readout for mitochondrial MMP. In the mitochondria with high MMP, the local concentration of PKMDR can be very high at the IMM, triggering dye quenching, resulting in a lower fluorescence lifetime. On the contrary, in mitochondria with low MMP, PKMDR was relatively dilute, giving a lower fluorescence intensity and a longer fluorescence lifetime (Fig. 2b).

To test our hypothesis, we treated HeLa cells with FCCP (Carbonyl cyanide 4- (trifluoromethoxy)phenylhydrazone, mitochondrial oxidative phosphorylation uncoupler), antimycin A (ETC complex III inhibitor), rotenone (ETC complex I inhibitor) and oligomycin (ATP synthase inhibitor), resulting in a decrease (FCCP, antimycin A, rotenone) or in an increase (oligomycin) in MMP[30] (Fig. 2c, Supplementary Fig. 4). Fluorescence lifetime imaging of PKMDR, referred to as PKMDR-FLIM, shows a clear increase after FCCP, antimycin A, and rotenone treatment ($\Delta\tau = 0.56$ ns, 0.37 ns, 0.24 ns, respectively), which was in line with a decrease in MMP (Fig. 2d). Similarly, oligomycin treatment resulted in a reduced PKMDR-FLIM value ($\Delta\tau = -0.29$ ns). We also measured the lifetime of PKMDR in cells treated with high and low osmolarity solutions, which resulted in mitochondrial swelling and crumpling. If the PKMDR lifetime is sensitive to changes in membrane tension, we would expect to observe opposite trends in the FLIM changes following hypo-osmotic versus hyper-osmotic shock. However, both treatments resulted in increases in the fluorescence lifetime of PKMDR, supporting the hypothesis that alterations in osmotic pressure led to mitochondrial dysfunction and a subsequent decrease in MMP (Supplementary Fig. 5). To rule out the potential influence of other membranous organelles during the FLIM measurements, we isolated the mitochondria from mouse cardiomyocytes and performed time-lapse imaging experiments using purified mitochondria. Consistent with the cell-based experiments, a gradual increase of fluorescence intensity, along with the concomitant decrease of fluorescence lifetime of PKMDR, was recorded in purified mitochondria upon the addition of succinate (substrate for complex II). Finally, upon the addition of the FCCP decoupler, which dissipated the MMP and dye accumulation, a sharp increase in PKMDR lifetimes was recorded, along with the change in fluorescence intensity in the opposite direction (Fig. 2e, f, and Supplementary Movie 1). The MMP changes during succinate and FCCP treatments were confirmed using TMRM stain and intensity imaging (Supplementary Fig. 6). The time-lapse imaging results using isolated mitochondria can rule out possible biochemical changes to a maximum extent, therefore strongly supporting that the fluorescence lifetime of PKMDR can directly report on MMP in live cells.

## PKMDR-FLIM can report mitochondrial respiration

The MMP is primarily generated by the ETC in the IMM. Electrons move through the ETC, driving protons from the matrix into the intermembrane space, creating a proton gradient. This gradient establishes the MMP, essential for ATP synthesis via mitochondrial F1FO ATP synthase. MMP magnitude indicates mitochondrial health and metabolic state (OXPHOS); a higher membrane potential suggests more active metabolism, whereas a reduced potential indicates mitochondria respiration slowdown or mitochondrial dysfunction. We therefore tested PKMDR-FLIM as an indicator of mitochondrial activity. The increase of mitochondrial activity via nutrient starvation is a canonical cellular response[31–33]. Depriving key nutrients like pyruvate and glutamine increases the levels of OXPHOS, which has been used as a stimulus to activate mitochondria[31–33]. In our test, nutrient starvation caused a decrease in PKMDR lifetime (Fig. 3a, b, Supplementary Fig. 7). Mitochondrial respiration and ATP production increased during the first two hours of serum starvation[34]. In line with this, PKMDR lifetime also decreased in the FBS starvation experiment (Fig. 3a, b, Supplementary Fig. 7). In contrast to starvation, senescent cells undergo various alterations in terms of their mitochondria's function, structure, and dynamics. Senescent cells have reduced mitochondrial membrane potential and lower ratios of ATP/ADP and NAD+/NADH[35,36]. We induced senescence in human umbilical vein endothelial cells (HUVEC) using the DNA-damaging agent doxorubicin[37]. Senescence was confirmed by senescence-associated β-galactosidase (SABG) staining and by measuring the expression of senescence-associated genes p16 and p21 by RT−qPCR (Supplementary Fig. 8). In senescent HUVECs, PKMDR-FLIM indicated increased fluorescence lifetimes, corroborating the reduced MMP (Fig. 3a, b).

Beyond the starvation and senescence models, the regulation of proteins in the mitochondrial respiratory chain also impacts the state of OXPHOS. We therefore tested if PKMDR-FLIM is responsive to OXPHOS deficiency, specifically a change in ETC complexes. NDUFS2, a core subunit of mitochondrial complex I, was essential for acute oxygen-sensing, development, and metabolism[38]. COX4I1, a common isoform of the COX4 subunit, was responsible for regulating COX and catalyzing the last electron transfer step in the respiratory chain of the mitochondria, which was the primary oxygen consumer (Fig. 3c)[39]. Genetically inactivating complex IV and complex I by knocking out its subunit COX4I1 and NDUFS2 in 143B cells[40] would both reduce respiration, as characterized by the elevation of PKMDR lifetimes (Fig. 3d, e, Supplementary Fig. 9).

The mitochondrial metabolism within a cell undergoes dynamic alterations across various stages of its lifecycle, manifesting as distinct metabolic profiles that are finely tuned to meet the specific energetic and biosynthetic demands of each phase. We next challenged whether PKMDR-FLIM can track mitochondrial metabolic dynamics throughout different stages. Mitochondria in early oocytes exhibit lower ETC activity compared to those of neighboring granulosa cells, and mitochondria metabolism increases during oocyte maturation[41,42]. Consistent with the fluorescence intensity imaging results of the TMRM (Supplementary Fig. 10), we found that PKMDR-FLIM can accurately map the mitochondria respiration of oocytes at different stages together with the granulosa cells (Fig. 3f–i). Meanwhile, it was well known that mitochondrial metabolism was accelerated after T cell

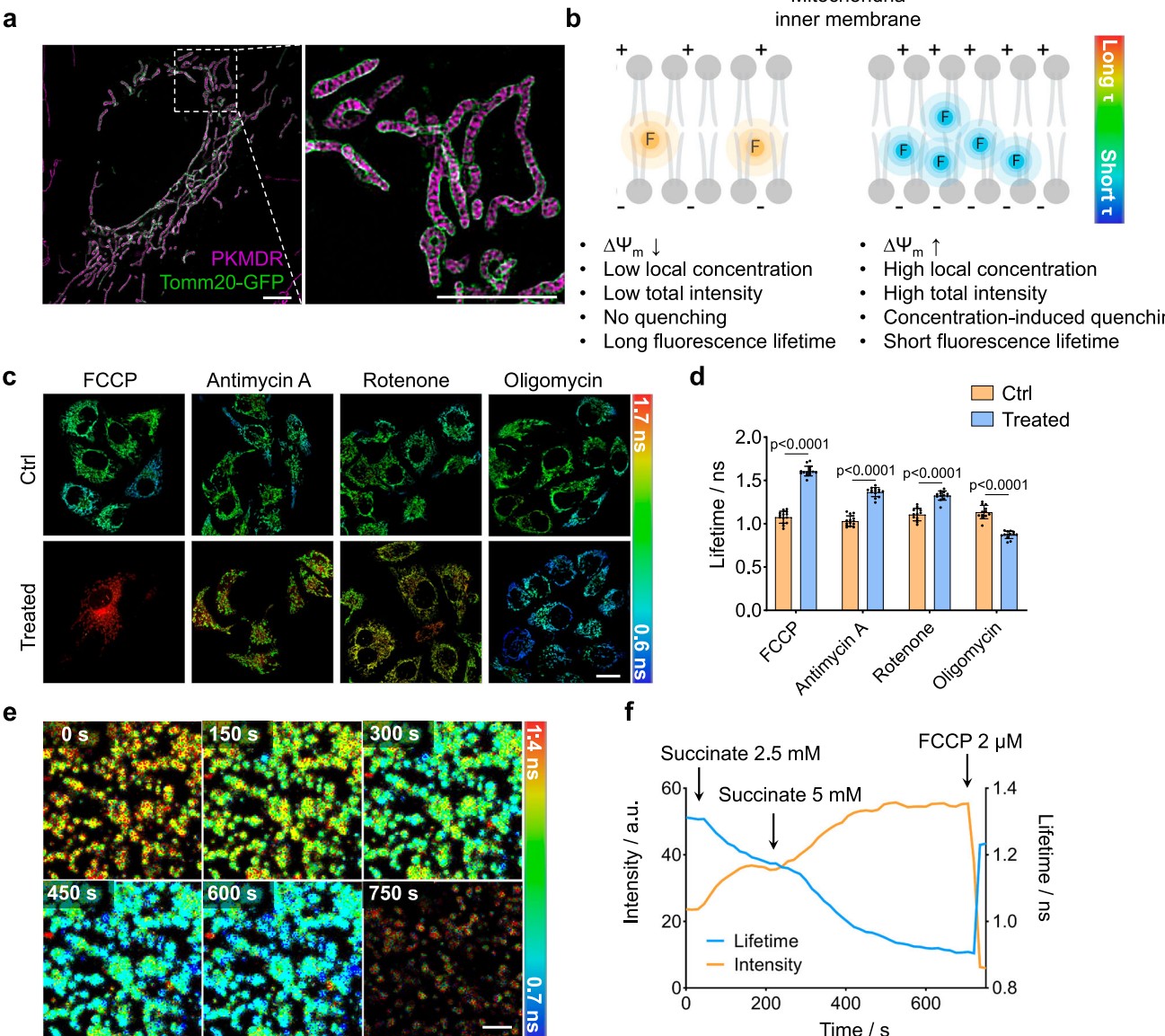

**Fig. 2 | PKMDR fluorescence lifetime is correlated to mitochondria inner-membrane potential in live cell imaging. a** SIM imaging of U2OS cells expressing Tomm20-GFP (green) stained with PKMDR (magenta) showcasing the inner-membrane localization of PKMDR. Scale bar, 5 μm. **b** Schematic illustration of PKMDR responding to the mitochondria membrane potential ($\Delta\Psi_m$). Under high $\Delta\Psi_m$, local PKMDR concentration was higher at the inner membrane, which shortens the fluorescence lifetime. Conversely, PKMDR is relatively diluted and dispersed in mitochondria at low $\Delta\Psi_m$, resulting in a reduced fluorescence intensity yet a longer fluorescence lifetime. **c** FLIM images of HeLa cells treated with FCCP (OXPHOS uncouplers), antimycin A (complex III inhibitor), rotenone (complex I

inhibitor), and oligomycin (ATP synthase inhibitor). Scale bar, 20 μM.
**d** Corresponding bar plots showing the average fluorescence lifetime change of mitochondria treated with FCCP ($n = 14$ cells), antimycin A ($n = 14$ cells), rotenone ($n = 13$ cells) or oligomycin ($n = 13$ cells). Data were presented as the mean ± SEM. P-values were calculated using two-tailed unpaired Student's t-test. **e** Timelapse FLIM imaging of purified mitochondria treated with succinate (two additions) followed by FCCP (2 μM). Images were acquired every 15 s. Scale bar, 5 μm.
**f** Fluorescence intensity (orange line) and fluorescence lifetime (blue line) over time, as measured in (**e**).

activation. PKMDR-FLIM can also profile such a change in mitochondria, where the naive T cells had a longer lifetime compared to activated T cells, which exhibited a shorter fluorescence lifetime (Fig. 3j, k). Overall, in starvation and senescence, ETC complex protein deficiency, oocyte development, and T cell activation models, PKMDR-FLIM was a robust protocol for direct visualization of mitochondrial metabolic status across various samples.

## PKMDR-FLIM unveils mitochondrial metabolic heterogeneity in tissues and developing embryos
Within the complex architecture of tissues, differentiated cells exhibit diverse metabolic specialization. Thus, the study of mitochondria

metabolic heterogeneity facilitates a deeper understanding of the development of various diseases. Previous membrane potential probes, such as TMRM, are subject to inaccuracies in fluorescence intensity imaging of tissues due to background fluorescence, light scattering, and variations in the path length of emitted light. Additionally, the fluorescence intensity signals can be compromised by uneven tissue sample thickness and defocusing, which may not accurately reflect the genuine changes in biological processes. Therefore, we explored whether PKMDR can label mitochondria in tissue samples and perform fluorescence lifetime imaging. As shown in Fig. 4a, high-quality images of mitochondria in brain organoids and tumor tissue can be recorded. The far-red fluorescence of PKMDR rendered

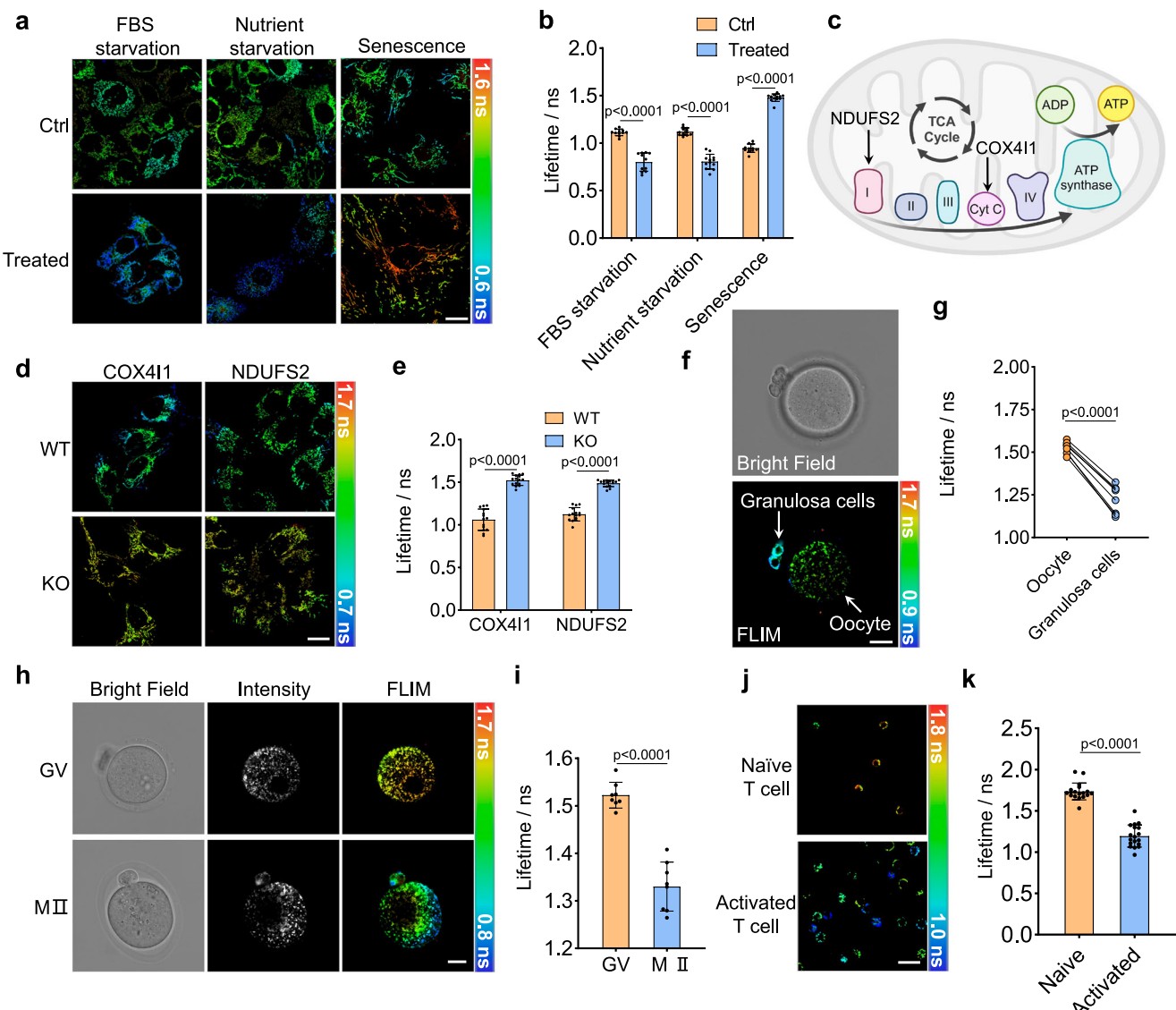

**Fig. 3 | PKMDR-FLIM signal indicates the activity of mitochondrial oxidative phosphorylation. a** PKMDR-FLIM images of HeLa cells after FBS starvation and nutrient starvation (cultured in DMEM lack of glucose, sodium-pyruvate, and glutamine), and HUVECs at senescence conditions. Scale bar, 20 µm. **b** Plots showing the average mitochondrial fluorescence lifetime under FBS starvation ($n = 11$ cells), nutrient starvation ($n = 12$ cells), and senescence HUVECs ($n = 11$ cells). **c** Schematic illustration of mitochondrial electron transport chain proteins. **d** PKMDR-FLIM images of wild type (WT), COX4I, and NDUFS2 knockout (KO) 143B cells. **e** Plots showing the average mitochondrial fluorescence lifetime in wild type ($n = 15$ cells) and knockout cells ($n = 15$ cells). **f** Bright field and FLIM images of PKMDR-stained mitochondria in germinal vesicle (GV) stage oocyte and granulosa cells. **g** Plots showing the average mitochondrial fluorescence lifetime in oocyte and granulosa cells ($n = 8$ cells). **h** Bright field, intensity and FLIM images of PKMDR-stained mitochondria in GV and metaphase II (MII) oocyte. **i** Plots showing the average mitochondrial fluorescence lifetime in GV and MII oocyte (n = 19 cells). **j** FLIM images of PKMDR-stained mitochondria in naïve and activated T cells. **k** Plots showing the average mitochondrial fluorescence lifetime in naïve and activated T cells ($n = 18$ cells). Statistical analysis was performed with unpaired two-tailed Student's t-test for (**b**, **e**, **i**, **k**), paired two-tailed Student's t-test for (**g**). Data were presented as the mean ± SEM. Scale bars = 20 µm.

compatibility for multiplexing with other fluorescent proteins or dyes. The strong fluorescence signal and the robust staining protocol of PKMDR allowed us to map the MMP heterogeneity at tissue level. At the same time, PKMDR-FLIM can clearly distinguish MMP differences between cells in brain organoids and tumor tissue (Fig. 4a).

We further investigated MMP in early-stage mouse embryos. In developing embryos, mitochondria play a crucial role, with their distribution, activity, and structure being closely tied to cellular functions[43,44]. From the two-cell to blastocyst stage, mouse embryos were approximately 50 to 150 µm. In these samples, intensity imaging of mitochondrial MMP using a TMRM probe was no longer accurate, as

the light scattering, refraction, and de-focusing in thick samples will complicate the signal analysis (Fig. 4b). Yet, PKMDR-FLIM suggested that from the 4-cell stage onward, MMP began to exhibit a heterogeneous pattern across different cells (Fig. 4b, c). It has been reported that certain pluripotency regulators were highly heterogeneously expressed among blastomeres of the 4-cell embryo[45–48], suggesting the presence of factors that modulate mitochondrial metabolic activity in individual blastomeres. Overall, PKMDR-FLIM offers direct visualization methods of MMP with spatial precision in tissues. The absolute-reading of FLIM is particularly advantages over intensity-based methods in imaging thick and scattering samples to avoid signal distortion.

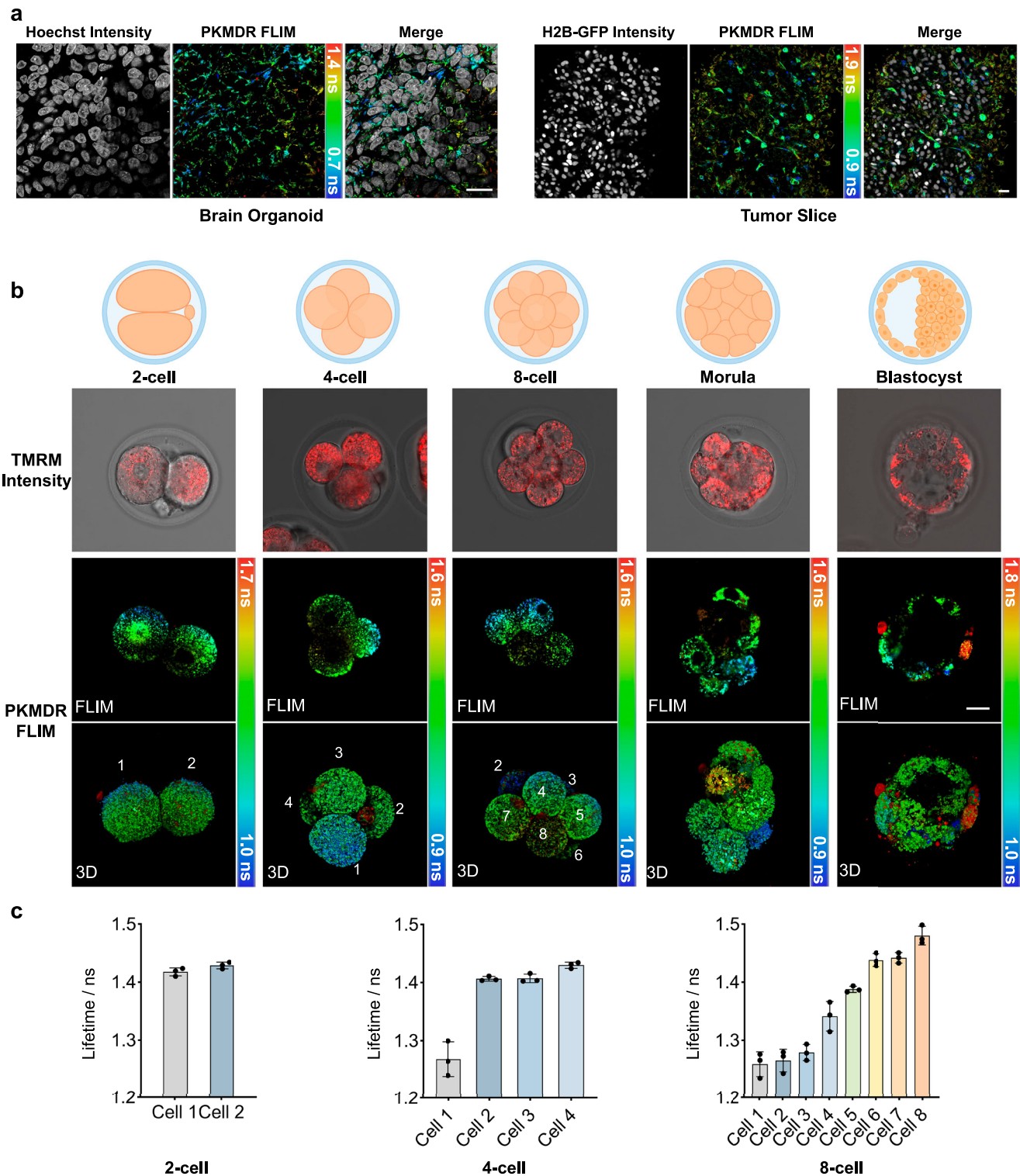

**Fig. 4 | PKMDR-FLIM maps MMP heterogeneity from cell to tissues. a** Intensity and FLIM images of PKMDR-stained mitochondria and nucleus (intensity images, grey) in brain organoid and tumor slice. **b** Bright field, intensity and FLIM images of TMRM- and PKMDR-stained mitochondria in mouse 2-cell, 4-cells, 8-cells embryos, morula and blastocyst. Schematic illustration of samples imaged in this figure. **c** Bar plots showing the average mitochondrial fluorescence lifetime in each cell ($n = 3$ samples). Scale bar = 20 μm. Data were presented as the mean ± SEM.

## Mitochondria in a single cell exhibit temporal and spatial heterogeneity

Mitochondrial metabolism was a dynamically changing process in both space and time. Yet, studying such heterogeneity heavily hinges on fluorescence imaging as it offers subcellular resolutions. Using PKMDR-FLIM, we investigated the distribution of mitochondrial membrane potential in various cell types. During early embryo development, the distribution and membrane potential of mitochondria were stage-specific and exhibited heterogeneity not only between cells but also within individual cells[49–51]. PKMDR-FLIM suggested that mitochondria with high MMP were distributed in the cell periphery, whereas low-MMP mitochondria were located in the central region of

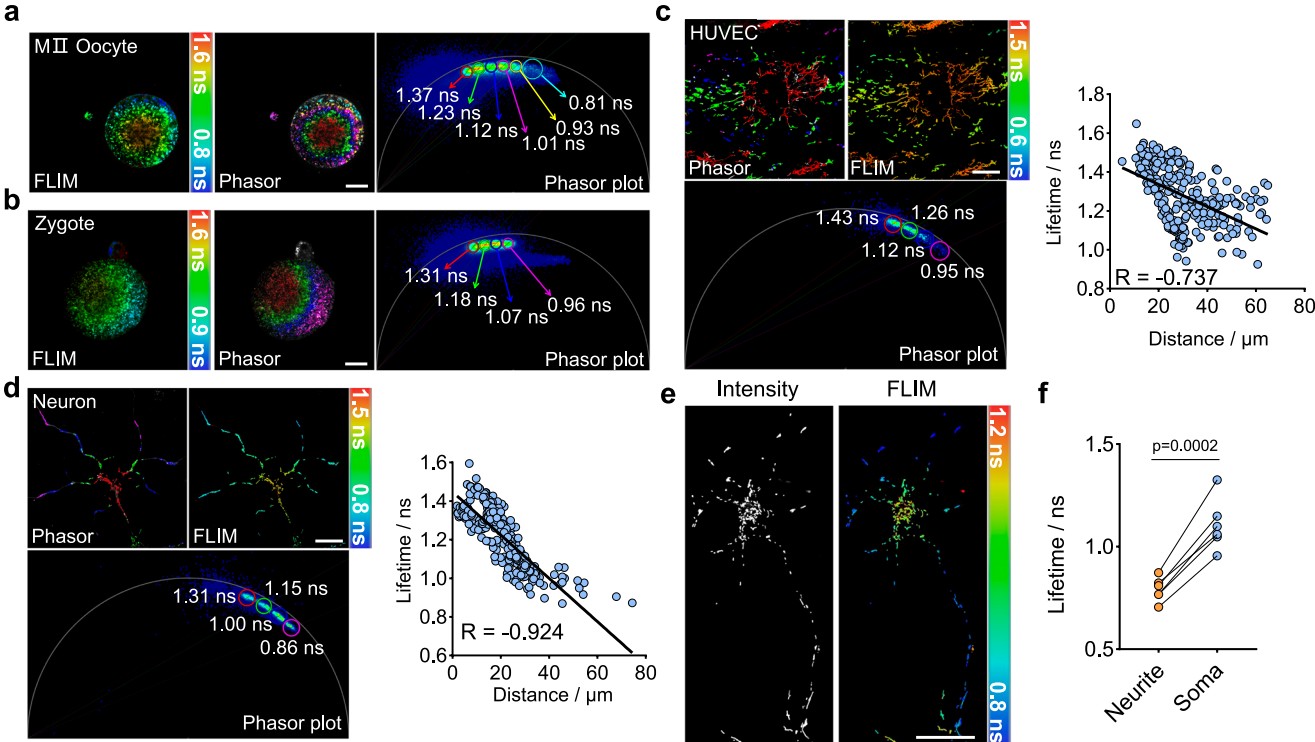

**Fig. 5 | Spatial heterogeneity of mitochondria is general in cells. a–d**, FLIM imaging and phasor plot of PKMDR-stained mitochondria in MII oocyte (**a**), zygote (**b**), senescent HUVEC (**c**) and neuron (**d**). Plots (**c**, **d**) showing correlation between mitochondrial fluorescence lifetimes and their distance to the center of cell.

R-values were calculated using Spearman correlation test. **e**, **f** Intensity and FLIM images of PKMDR-stained mitochondria in neuron. Plots (**f**) showing the average fluorescence lifetime observed in neurite and soma (*n* = 6 cells). P-values were calculated using paired two-tailed Student's t-test. Scale bars, 20 μm.

MII oocytes (Fig. 5a). Such a distribution pattern was also found in zygotes (Fig. 5b). Notably, the spatial distribution patterns of MMP in oocytes remain a subject of ongoing debate. Conflicting results have been reported between TMRM measurements (showing MMP gradient decreasing from the central region to the periphery) and those obtained through JC-1 imaging (demonstrating an inverse peripheral-to-central gradient)[52–55]. Our analysis aligns closely with the spatial patterns observed in JC-1, which providing support and insights for the study of spatial heterogeneity of MMP.

We also mapped the MMP in HUVEC and neurons. In these spreading cells, mitochondrial membrane potential increased from the nuclear region to the cell periphery (Fig. 5c, d). These results highlight how mitochondrial membrane potential varies spatially. Furthermore, the FLIM analysis of mitochondria in different regions in neurons showed decreased levels of MMP in the neuronal soma compared to terminals (Fig. 5e, f). This result was in line with a previous study, where the metabolic flux and cellular respiration analyses of isolated neuronal somata and synaptosomes evidence suggested that the neuron soma exhibit increased levels of aerobic glycolysis and decreased levels of OXPHOS compared to the terminals, both in their resting and activated states[56]. Therefore, using PKMDR-FLIM, the heterogeneity of MMP at the sub-cellular level can be unambiguously imaged, supplementing and supplanting other methods for studying metabolism with lower spatial and temporal resolutions.

Finally, we demonstrated that the extraordinarily low phototoxicity of PKMDR enables time-lapse FLIM recording, revealing the temporal regulation of mitochondrial MMP. The long stretch of video captures multiple mitochondrial events with sudden decreases in membrane potential followed by recoveries (Fig. 6a, b and Supplementary Movie 2). These events were reminiscent of the well-studied Mito-Flash events, which were optical readout systems that use frequency coding to represent free-radical generation and energy

metabolism processes within individual mitochondria[57,58]. In primary neurons, PKMDR-FLIM can record the real-time transport of mitochondria between neurites and the soma, simultaneously revealing both their location and membrane potential (Fig. 6c, Supplementary Movie 3). The complexity of mitochondrial trafficking in neurons has remained difficult to study due to the limitations of traditional imaging techniques, and the relationship between mitochondrial transport and metabolism has long been a subject of debate[59–62]. PKMDR-FLIM holds the potential to address key questions regarding mitochondrial function and its role in neuronal health and disease.

## Discussion

From a spectroscopy point-of-view, this work established the generality of concentration-dependent fluorescence lifetime change of cationic dyes using measurements in solution, in GUV, and in live cells. At higher dye concentrations, the non-linear behavior, such as aggregation and non-radiative energy transfer, begins to prevail, giving a decreasing fluorescence lifetime. Notably, each tested dye exhibits a distinct "critical concentration" for lifetime decay, which was correlated with the hydrophobicity and self-aggregation tendency of the dye. Consequently, TMRM and TMRE, the relatively hydrophilic rhodamine dyes, give smaller lifetime responses at only very high concentrations, while PKMDR, as a hydrophobic cyanine dye, exhibits a proper dynamic range for FLIM imaging in the mitochondrial context. Fluorescent lifetime can become a parameter for studying the fluorophore interactions at high concentrations. At the same time, this work adds another direction to FLIM imaging beyond measuring microenvironmental factors such as viscosity, pH, polarity, and FRET.

From the probe and method perspective, PKMDR, originally devised for structural imaging of IMM, can now be used in conjunction with FLIM for MMP imaging, supplementing and supplanting traditional MMP imaging techniques such as TMRM intensity imaging and JC-1

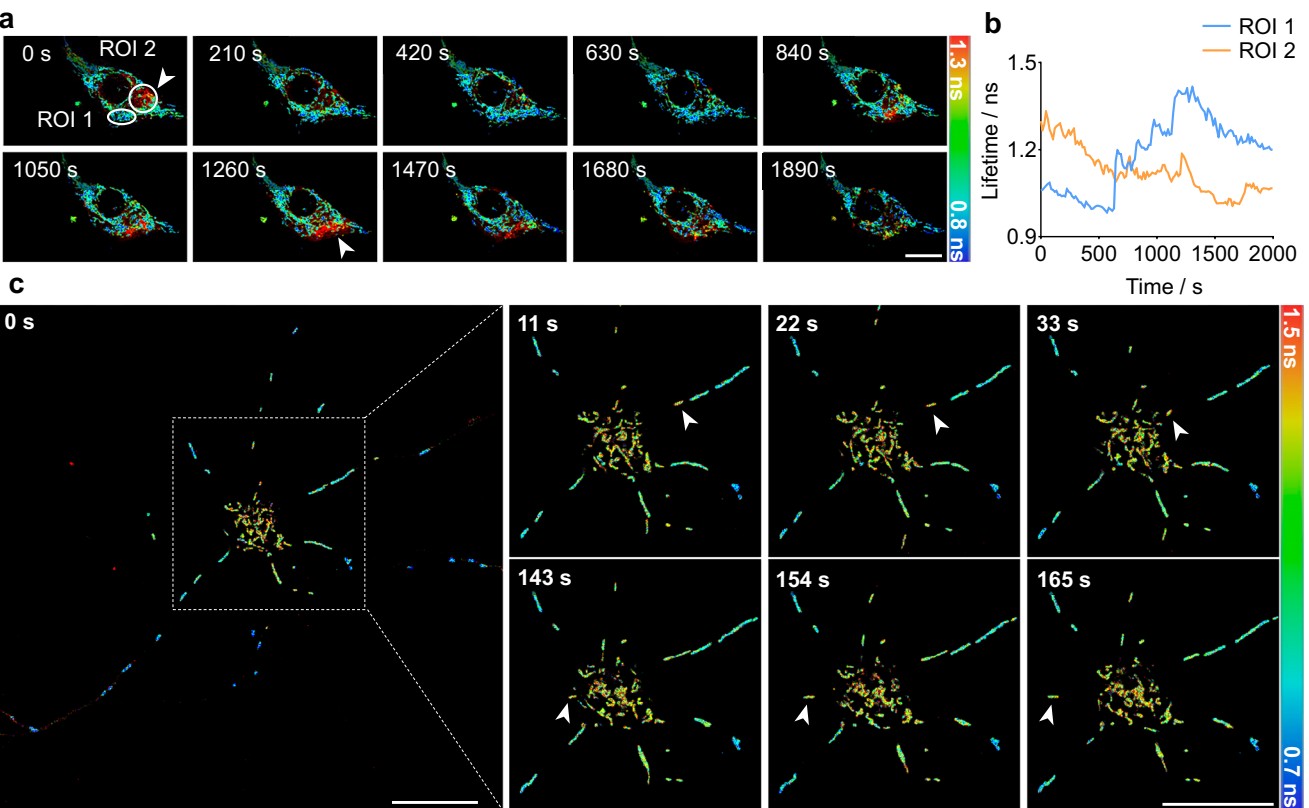

**Fig. 6 | Time-lapse FLIM recording reveals the temporal heterogeneity of mitochondria. a** Time-lapse FLIM imaging of PKMDR-stained mitochondria in L929 cells. Images were acquired every 15 s. Spontaneous mitochondrial membrane potential change events were indicated with arrows. **b** Fluorescence lifetime profiles of the two ROI specified in (**a**). **c** Time-lapse FLIM imaging of mitochondria in a neuron. Images were acquired every 11 s. Two mitochondria underwent transportation are indicated with arrows. Scale bars, 20 μm.

ratiometric imaging. Compared to the intensity signal of TMRM, PKMDR's low phototoxicity eliminates fluorescence decay artifacts during prolonged imaging. Moreover, PKMDR lifetime exhibits minimal Z position variation (1.5% coefficient of variation vs. TMRM intensity's 16% coefficient of variation), enabling reliable 3D quantification in specimens like embryonic blastomeres at different depths where TMRM becomes unreliable (Supplementary Fig. 11). The straightforward staining protocol of PKMDR, the bright fluorescence and minimal phototoxicity, and the direct reading via a FLIM microscope collectively give a strong competitive edge over protocols based on JC-1 ratiometric imaging. As we demonstrate that several dye candidates can be exploited for FLIM imaging, PKMDR gives strong signal, high dynamic range, and long time-lapse recording, offering the best practice.

For mitochondrial biology, the spatial and temporal heterogeneity of MMP was clearly recorded with PKMDR-FLIM in multiple primary cells, organoids, and developing embryos, with subcellular resolutions over a long time. Notably, the insights we acquired from imaging data, such as the different mitochondrial metabolism in neurite and soma, corroborates with biochemical assays in the literature[56]. A possible future direction is to correlate the PKMDR-FLIM signal with additional fluorescent protein-based metabolite sensors, giving multi-color imaging of mitochondrial parameters that offers panoramic view of cellular metabolism. Overall, PKMDR-FLIM offers unique insights into mitochondrial metabolism, particularly in spatially and temporally sophisticated samples such as neurons, oocytes and developing embryos.

PKMDR-FLIM has certain limitations. First, the prolonged data acquisition time required for time-domain FLIM imaging, which necessitates sufficient photon accumulation for accurate lifetime calculations, inherently restricts temporal resolution. This is a particular issue at low ΔΨm where PKMDR's fluorescence intensity becomes lower. Another limitation is that PKMDR's concentration-dependent fluorescence lifetime behavior imposes stringent experimental constraints. The probe's lifetime-concentration correlation window demands a critical threshold concentration, which can be achieved only under an optimized staining protocol outlined in our study. For non-standardized biological models, preliminary dose-response experiments are essential. While these limitations are inherent to PKMDR's design, our methodology incorporates rigorous controls and validation steps to mitigate their impact on data interpretation.

Here we showcase that the concentration-dependent fluorescence lifetime change, previously regarded as a non-preferred spectroscopic phenomenon, can be leveraged in imaging of mitochondrial membrane potential. PKMDR, with high brightness, far-red emission, extraordinarily low phototoxicity, and proper aggregation tendency, serves as an ideal probe for FLIM mapping of the heterogenous mitochondrial metabolism in space and time. With a handy protocol, strong signal, and versatile sample scope, PKMDR-FLIM could pave the way for unveiling mitochondrial metabolism over time in physiological and pathological conditions.

## Methods
### Fluorescence microscopy
Confocal laser scanning microscopes (TCS SP8 or STELLARIS 8 FALCON; Leica) with a pulsed white light laser (WLL) fitted with an APO 100×/1.40 oil and an APO 40×/1.32 oil objective lens, as well as HyD detectors, were used for confocal and FLIM imaging. FLIM images were

acquired under excitation at 633 nm (SP8, WLL, 80 MHz) or 638 nm (STELLARIS 8, WLL, 80 MHz) and the emission was collected in the range of 650–700 nm. The pinhole was set to 1.0 AU. Data were collected using LAS X (Leica) at 512 × 512 or 1024 × 1024 pixel resolution and frame accumulations of 1–10 times, or collecting 500 photons per pixel (Supplementary Table 2). FLIM data were analyzed using the LAS X software (Leica Microsystems) by fitting a mono-, bi-, or triexponential decay model (n-exponential reconvolution, Eq. 1) to the decay ($\chi^2 < 1.3$, Eq. 2) (Supplementary Table 2). Intensity-weighted lifetime (Eq. 3) was used as mean fluorescence lifetime in each image. The instrument response function (IRF) used for deconvolution was measured under the same microscope settings. For time-lapse imaging, multiple images were acquired with a time delay of 11 s or 15 s interval between the starting point of capturing two successive images. Phasor plot analysis in Fig. 5 is performed in LAS X software (Leica Microsystems). After selecting a region of interest around the cell for analysis, the corresponding phasor plot is displayed, and pixels with similar lifetimes are grouped into clusters. Distinct clusters emerge when MMPs in an individual cell are different. Pixels forming each cluster can be identified in the lifetime image and thus separated. No threshholding is applied.

$$y(t) = \left\{ IRF\left(t + Shift_{IRF}\right) + Bkgr_{IRF} \right\} \bigotimes \left\{ \sum_{i=1}^{n-1} A[i] e^{\left(-\frac{t}{\tau[i]}\right)} + Bkgr \right\} \quad (1)$$

n: Number of exponential components
 A: Amplitudes – Exponential pre- factors
 τ: Exponential Decay Times (e.g. lifetimes)
 Bkgr: Tail Offset – Correction for background
 Shift_{IRF}: IRF Shift – Correction for IRF displacement
 Bkgr_{IRF}: Irf Background – Correction for IRF background

$$\chi^2 = \sum_{k=1}^{n} \frac{\left[N(t_k) - N_c(t_k)\right]^2}{N(t_k)} \quad (2)$$

$N(t_k)$: Measured fluorescence decay function
$N_C(t_k)$: Calculated decay function
n: Evaluated across the number of data points

$$\tau_{Av\ Int} = \frac{\sum_{k=0}^{n-1} I[k]\,\tau[k]}{I_{sum}} \quad (3)$$

I: Intensities associated with each exponential component, normalized to the time resolution of the measured decay curve in order to be displayed in photon counts
 $I_{Sum}$: of Fluorescence Intensity for all components
 $\tau_{Av\ Int}$: Mean Photon Arrival Time – Intensity weighted average lifetime

HIS-SIM imaging (Fig. 2a) was performed by following the previous report[63]. A commercial structured illumination microscope (HIS-SIM) was used to acquire and reconstruct the cell images. To further improve the resolution and contrast in reconstructed images, sparse deconvolution was used by following the previous report[64].

## Cell culture and treatment

HeLa, U2OS, MC-38, 143B and L929 cell lines were grown in DMEM medium with 10% v/v fetal bovine serum (FBS, Vistech), 1% v/v penicillin-streptomycin mixture (Gibco) and GlutaMax™ (Gibco) supplement addition. Regarding that COX4I1-KO 143B and NDUFS2-KO 143B cell lines had conspicuous attenuation in oxygen consumption rate, we added extra 3 mM pyruvate and 50 μg/mL uridine in aforementioned DMEM system to sustain the cell viability for these two gene knocked-out cell lines. HUVEC were cultured in endothelial cell medium (ScienCell) with 5% FBS (ScienCell), 1% ECGS (ScienCell) and 1% P/S (ScienCell). All cells were incubated at 37 °C with 5% $CO_2$. For

regulating mitochondrial membrane potential (Fig. 2c-d), FCCP (2 μM), antimycin A (10 μM), rotenone (10 μM), oligomycin (20 μM) were used for modulation of mitochondrial membrane potential in live cells.

For induction of senescence (Fig. 3a, b), HUVEC cells were treated with 200 nM of doxorubicin for 48 h, and then the culture medium was refreshed. Cells were maintained in culture and imaged at days 2 after treatment.

Hypertonic and hypotonic treatment (Supplementary Fig. 5) were performed using a previously published protocol[21]. Hypotonic solutions was salt-free HANKS buffer (140 mM NaCl, 2.5 mM KCl, 0.5 mM $MgCl_2$, 1.2 mM $CaCl_2$, 5 mM glucose, 10 mM HEPES pH 7.5), and supplemented with 500 mM of NaCl was used to make hypertonic solutions. Cell starvation model was established using a previously published protocol with modifications[21]. For FBS starvation (Fig. 3a, b), cells were cultured in FBS-free media for 2 h before imaging. For nutrient starvation (Fig. 3a, b), cells were cultured in DMEM media without glucose, sodium-pyruvate and glutamine (Thermofisher Scientific) for 2 h before being taken for imaging.

## PKMDR labeling for live-cell FLIM imaging

Cells were seeded into a glass-bottom dish (Cellvis) a day before imaging with the cell density at around $4 \times 10^4$ cells/mL. For labeling of mitochondria, cells were stained with DMEM supplemented with 250 nM PKMDR for 20 min at 37 °C. Following the staining procedure, the cells were washed up to three times with pre-warmed PBS or culture medium to remove the unbound dye, followed by the addition of DMEM for live cell imaging. If necessary, further incubation at 37 °C for 15 min before imaging can reduce unspecific ER labeling. For multicolor labeling, we recommend sequential staining if the used dyes require different incubation times.

## Lentiviral production and generation of stable cell lines

pBOB-EF1-H2B-GFP plasmid, psPAX2 and pMD2.G were kindly offered by Dr. Jing Hu's lab (Peking University). Lentivirus production was carried out in HEK293T cells cultured to 90–95% confluency. The cells were co-transfected using Lipo8000™ Transfection Reagent (Beyotime) with a mixture of lentiviral vectors (containing the cDNA of interest), along with the packaging plasmids psPAX2 and pMD2.G at a ratio of 10:7:3. Virus-containing supernatants were collected and filtered through 0.45 μm filters 48 h post-transfection, then employed to infect target cells for 24 h in the presence of 10 μg/ml polybrene (Beyotime). Infected cells were selected by 10 μg/ml puromycin (Beyotime). H2B-GFP expression was validated by confocal imaging (Fig. 4a).

## Tumor tissue imaging

All animal procedures were approved by the Peking University Animal Use and Care Committee and complied with the standards of the Association for Assessment and Accreditation of Laboratory Animal Care (code number IMM-ChenZX−1). For cancer models (Fig. 4a), female C57BL/6 J mice (8 weeks) (Beijing Vital River Laboratory Animal Technology) were inoculated subcutaneously with $1 \times 10^6$ MC38 H2B-GFP cells into the right flank. After tumor growth up to 100 mm³, tumors were rapidly removed and placed in cold (4 °C) HBSS for 1 min. Next, tumors were dissected for mounting to the cutting stage, and 100 μm thick slices were cut in ice-cold HBSS using a VT1200 vibratome (Leica). The slices were stained with DMEM supplemented with 500 nM PKMDR for 45 min at 37 °C. After wash for 3 times, the slices were placed to glass-bottom dishes (Cellvis) and imaged using TCS SP8 FALCON confocal microscope (Leica).

## Sen-β-Gal activity

Cells were fixed for 15 min in 2% PFA at room temperature, and then stained for 16 h at 37 °C in the dark using the Senescence β-Galactosidase Staining Kit (Beyotime). After incubation, cells were

imaged using a Zeiss Axiovert 200 microscope (Zeiss) and analyzed with ImageJ (Supplementary Fig. 8).

## qPCR

Total RNA (1 µg) was extracted from cells with RNAsimple Total RNA Kit (TIANGEN) and reverse-transcribed into cDNA by HiScript III 1st Strand cDNA Synthesis Kit (Vazyme). The cDNAs were amplified by the TB Green® Premix Ex Taq™ II (Takara) using the CFX96TM Real-Time System (Bio-Rad). $2^{-\Delta\Delta Ct}$ method was used to quantify the relative expression. The mRNA levels were normalized by GAPDH. The primer sequences are as follow: (1) 5'-GAT CCA GGT GGG TAG AAG GTC-3' as p16-F and 5'-CCC CTG CAA ACT TCG TCCT-3' as p16-R; (2) 5'-TGT CCG TCAG AAC CCA TGC-3' as p21-F and 5'-AAA GTC GAA GTT CCA TCG CTC-3' as p21-R; (3) 5'-GGA GCG AGA TCC CTC CAA AAT-3' as GAPDH-F and 5'-GGC TGT TGT CAT ACT TCT CAT GG-3' as GAPDH-R (Supplementary Fig. 8).

## Fluorescence intensity and lifetime measurement

PKMDR, TMRM, and TMRE were dissolved in DMSO for fluorescence intensity and fluorescence lifetime measurement (Fig. 1b–d, Supplementary Fig. 2). 10 mM stock solution was diluted with DMSO to different concentration ranging from 10 nM to 5 mM for each dye. Fluorescent intensity curve was acquired by TECAN Infinite® 200 Pro Multimode Microplate Readers, where 384-well microplate was used to hold diluted samples for fluorescence scanning at room temperature. Fluorescent lifetime measurement was conducted using FLS980 Spectrometer (Edinburgh Instruments) containing an integrating sphere at room temperature with in a 1 cm square quartz cuvette. Fluorescence lifetime data were analyzed using the Fluoracle software (Edinburgh Instruments) by fitting a mono-exponential decay model (n-exponential reconvolution) to the decay. The instrument response function (IRF) used for deconvolution was measured under the same instrument settings. Measurements were carried out at their excitation wavelength (PKMDR, 644 nm; TMRM and TMRE 515 nm).

## Measurement of ROS generation of fluorophores

Singlet oxygen quantum yields of TMRM, JC−1, Mitorotor-1, SiRM, and PKMDR were measured with 1,3-Diphenylisobenzofuran (DPBF, Dibai), which can combine with singlet oxygen as a decaying fluorescent indicator. Absorbances at 525 nm of TMRM, JC-1, and Mitorotor-1 solution are modulated to 0.11 in air-saturated ACN where TMRM played as a reference. Absorbances at 625 nm of SiRM and PKMDR solution are modulated to 0.11 in air-saturated ACN in which SiRM played as a reference. Treated the system with DPBF (final concentration $8 \times 10^{-5}$ M), then irradiated TMRM, JC-1, and Mitorotor-1 with 520–530 nm LED lamp (50 mW/cm², SiRM and PKMDR with 620–630 nm LED lamp). The slope of linear decay of DPBF absorption at 415 nm is proportional to the absolute singlet oxygen quantum yield (Supplementary Table 1).

## GUV preparation and imaging

1,2-Dioleoyl-sn-glycero-3-phosphocholine (DOPC) (Fig. 1e–j) were used as lipids for the preparation of giant unilamellar vesicles (GUVs), which is dissolved in CHCl₃ to a final concentration of 10 mM. GUVs were prepared using electro formation technique by Vesicle Prep Pro (Nanion Technologies). To immobilize lipid vesicles for confocal imaging, GUVs solution supplemented with different concentration of PKMDR was mixed with 0.5 wt% low gelling temperature agarose (Sigma) solution in 1:1 ratio. The mixture was dropped onto a glass slide and imaged with TCS SP8 FALCON confocal microscope (Leica).

## Isolation and imaging of cardiac mitochondria

Mouse cardiac mitochondria (Fig. 2e, f) were isolated using a previously published protocol with modifications[65]. Glass-bottom dishes were coated with a poly-D-lysine solution (Gibco) and placed at 37 °C overnight. Female C57BL/6 J mice (8 weeks) (Beijing Vital River Laboratory Animal Technology) were anesthetized with an i.p. injection of tribromoethanol (500 mg/kg) and dissected. Heart was removed and washed with pre-cooled mitochondria isolation medium (300 mM sucrose, 10 mM HEPES (pH 7.2), 1 mM EGTA, and 0.5 mg/mL BSA, adjust pH to 7.2 with potassium hydroxide), minced, and homogenized on ice. Centrifuge the homogenate at 4 °C for 10 min at $600 \times g$. Then collect the supernatant and further centrifuge at 4 °C for 10 min at $6000 \times g$. Re-suspend the pellet and place them to poly-D-lysine coated glass-bottom dishes, then centrifuge at 4 °C for 6 min at $3220 \times g$. Replace the isolation medium to respiration buffer (300 mM sucrose, 10 mM KCl, 10 mM Tris-HCl, 10 mM K₂HPO₄, adjust pH to 7.2 with potassium hydroxide) at room temperature. After staining with 500 nM PKMDR, 2.5 mM of succinate or 2 µM FCCP was added to mitochondria in respiration buffer at the time of imaging for modulation of mitochondrial membrane potential.

## Oocyte and early embryo collection

To obtain oocytes or pre-implantation embryos (Figs. 3f–i, 4b, c, 5a, b), 3-week-old C57BL/6 N (10–12 g) female mice were intraperitoneally injected with pregnant mare's serum gonadotropin (PMSG, 10 IU) and human chorionic gonadotrophin (HCG, 10 IU). The zona pellucida was gently removed by treatment with Tyrode's solution (Merck Millipore, MR-004-D). Oocytes at GV and MII stages were collected at 0, 12 h post HCG. To collect GV oocytes, cumulus–oocyte complexes (COCs) was obtained by manually rupturing antral ovarian follicles, and cumulus cells were removed by repeatedly pipetting. To collect MII oocytes, COCs were harvested from the ampullae of the oviduct, and cumulus masses were removed by incubation in a hyaluronidase medium (Sigma, H4272). Pre-implantation embryos were cultured in KSOM media (Merck Millipore, MR-121-D) at 37 °C with 5% CO₂ and collected at the following time points after HCG stimulation: Zygote (21–23 h), two-cell embryos (43–45 h), four-cell embryos (54–58 h), eight-cell embryos (68–70 h), morula-stage (78–80 h) and blastocysts (96–100 h).

Oocytes and preimplantation embryos (zygote to morula) were treated with 500 nM PKMDR in pre-warmed KSOM medium for 20 min and blastocyte-stage embryos were treated for 1 h at 37 °C with 5% CO₂ subsequently washed three times in fresh KSOM medium. Images were taken with the STELLARIS 8 FALCON confocal microscope system (Leica).

## T cell isolation and culture

Murine CD8⁺ T cells (Fig. 3j, k) were isolated from the spleens of C57BL/6 J mice (8 weeks) by EasySep™ Mouse CD8⁺ T Cell Isolation Kit (STEMCELL Technologies) activated with Dynabeads® Mouse T-Activator CD3/CD28 beads (Thermofisher) for 48 h and maintained in RPMI 1640 medium (Gibco) with 10% FBS, 50 µM β-mercaptoethanol (Sigma-Aldrich) and 60 U of IL-2 (PeproTech). For labeling of mitochondria, naïve or activated CD8⁺ T cell transferred to glass-bottom dishes coated with poly-D-lysine and stained with RPMI 1640 medium supplemented with 250 nM PKMDR for 30 min at 37 °C, respectively.

## Neuron isolation and culture

For primary rat hippocampus neuron culture (Figs. 5d–f, 6c), glass-bottom dishes were coated with a poly-D-lysine solution (Gibco) and placed at 37 °C overnight. In order to isolate neurons, the skulls of neonatal Sprague-Dawley rats were cut off using scissors. Next, the brain was extracted from the skull and placed into a DMEM medium pre-cooled to 4 °C. Subsequently, the hippocampi were isolated from the brains using a dissecting microscope, fragmented into small segments, and then exposed to Trypsin-EDTA (0.25%) for a duration of 15 min at a temperature of 37 °C. Subsequently, the trypsin was

carefully substituted with preheated (37 °C) DMEM solution containing 10% FBS. Following a series of pipetting actions for a duration of 1 min and subsequent incubation on ice for 5 min, the tissue sections were then settled at the bottom of the centrifuge tube. The supernatant was collected and diluted with neural culture medium (Neurobasal™ medium supplemented with B-27™ supplement, GlutaMAX™ supplement, and penicillin-streptomycin) and then transferred to glass-bottom dishes coated with poly-D-lysine. Every four days, half of the neural culture medium was substituted with a new medium. For labeling of mitochondria or cell membrane and mitochondria, cells were stained with DMEM supplemented with 10 nM or 250 nM PKMDR for 20 min at 37 °C, respectively.

### Organoids
Cortical organoids (Fig. 4a) were generated using a previously published protocol with modifications[66]. In brief, human embryonic stem cells (hESCs) were cultured to 70%–80% confluent and incubated with Accutase (Invitrogen) at 37 °C for 5 min and dissociated into single cells. About 9000 live cells in 150 µl PGM1 medium (Cellapy) supplemented with CEPT Cocktail (GLPBIO) and 0.0125% PF-127 (Sigma) were added to each well of a 96-well non-treated round-bottom microplate (Corning) to form embryoid bodies. Twenty-four hours following cell aggregation (day 0), 3D hESC spheroids were treated with KSR medium supplemented with 10 µM SB-431542 (MedChemExpress), 2.5 µM dorsomorphin (Selleck) and 1.25 µM XAV-939 (MedChemExpress). For imaging, cortical organoids were collected at day 16, treated with 300 nM PKMDR and 5 µg/mL Hoechst 33342 in pre-warmed culture medium for 30 min.

### Mitochondrial segmentation and tracking
The image analysis platform was set up using a workstation with the following specifications was employed: a 3.70 GHz Intel i9-10900X CPU, an RTX TITAN NVIDIA GPU with 24 GB of memory, and 256 GB of RAM. The software platform used was based on Python 3.8. Additionally, certain algorithms relied on ImageJ[67] and TrackMate[68] for their implementation.

For the images acquired from the imaging system, we initially apply the median filtering and Rolling ball algorithms[69] to denoise. Subsequently, the denoised images are binarized using the Otsu algorithm[70] for mitochondrial segmentation. The morphological opening operations were introduced to eliminate small objects. Following this, label images were generated for each mitochondrial region. During mitochondrial tracking, the aforementioned mitochondrial image is input into TrackMate for Simple LAP Tracker analysis.

To measure the relationship between the fluorescence lifetime of each mitochondrion and its distance from the nucleus. We first manually labeled the position of the nucleus centroid for each cell. The average distance from the nucleus and the average fluorescence lifetime of all pixels within each mitochondrion were then calculated. Each mitochondrion constitutes a point on the scatter plot. Finally, the Spearman correlation coefficient was calculated (Fig. 5c, d).

### Statistical analysis
Statistical analysis and drawing of graphs were performed using Prism version 10.0 (GraphPad Software). Normality tests were used where appropriate to select statistical tests. Each figure legend indicates the statistical tests used as well as the threshold below which $p$-values are plotted. Data are displayed as scatter plots, with single points either representing biological replicates (study participants) or single cells, overlaid on bars + standard error of mean. All $t$-tests were performed as two-tailed tests.

### Ethics approval and consent to participate
All procedures were approved by the Peking University Animal Use and Care Committee and complied with the standards of the Association

for Assessment and Accreditation of Laboratory Animal Care (code number IMM-ChenZX-1).

### Reporting summary
Further information on research design is available in the Nature Portfolio Reporting Summary linked to this article.

## Data availability
All data associated with this study are presented in the main text or Supplementary Information. The raw numbers for charts and graphs are available in the Source Data. Source data are provided with this paper.

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

## Acknowledgements

This project was supported by funds from National Key R&D Program of China (2021YFF0502904) and the Beijing Municipal Science & Technology Commission (Project: Z221100003422013). We thank Prof. Heping Cheng, Prof. Xianhua Wang, Prof. Stefan Jakobs and Dr. Till Stephan for the helpful discussions. We thank Guangzhou CSR Biotech Co., Ltd for live-cell imaging by using their commercial super-resolution microscope (HIS-SIM), data acquisition, SR image reconstruction, analysis and discussion. Beijing National Laboratory for Molecular Sciences (grant 269 BNLMS202407 to Z.C.) and the International Cooperation and Exchange of the National Natural Science 270 Foundation of China (grant W2412031 to Z.C.). Figures 1a, 2b, 3c and 4b were created with BioRender.com.

## Author contributions

Z.C. conceived the study. D.S. designed, performed, and analyzed the assays and performed FLIM imaging. T.A. provided mouse embryo and oocytes. X.S. contributed to the phototoxicity and imaging experiments. Z.L. performed mitochondrial segmentation and tracking. L.R. and S.P. provided constructive suggestions. Y.L. performed the chemical synthesis and characterizations. S.F. and S.D. provided knockout cells and organoids, respectively. A.M., L.C., H.J., and Z. C. supervised the project. D.S. and Z.C. wrote the paper.

## Competing interests

Z.C. is the inventor of the patent of PKMDR (ZL202010492298.8), whose value may be affected by this paper. PK Mito DeepRed is commericially distributed by Genvivo and Spirochrome, where Z.C. and L.R. are their respective co-founders. The remaining authors declare no competing interests.
