## [Transparent Peer Review file · Nature Communications]

Imaging mitochondrial membrane potential via concentration-dependent fluorescence lifetime changes

Corresponding Author: Professor Zhixing Chen

Version 0:

Reviewer comments:

Reviewer #1

(Remarks to the Author)

This article presents a new approach to monitor mitochondrial membrane potential (MMP) using the PKMDR probe in combination with fluorescence lifetime imaging microscopy (FLIM). The study demonstrates that PKMDR's fluorescence lifetime serves as a sensitive readout for MMP, revealing mitochondrial metabolic heterogeneity in various cell types, organoids, and tissues. The authors provide a thorough literature review on existing mitochondrial imaging tools and their limitations, setting the stage for their new method. They explain the photophysical properties of PKMDR and how its fluorescence lifetime changes with concentration. The experimental validation, involving both in vitro and in vivo systems, is sufficient, demonstrating PKMDR-FLIM's utility under various metabolic conditions and in different cell types. The findings regarding metabolic heterogeneity within and between cells, as well as the temporal dynamics of mitochondrial membrane potential, are partially novel and have good implications for mitochondrial biology and related diseases.

However, some comments and suggestions for improvement could strengthen the manuscript:

1. The fluorescence lifetime of a dye is generally considered to be independent of its concentration, which represents a significant advantage over fluorescence intensity measurements, as it is not affected by variations in dye concentration within cells. However, the authors present a unique probe, PKMDR, whose fluorescence lifetime specifically depends on concentration to characterize mitochondrial membrane potential. This raises an important question: How can the authors ensure that the parameter being characterized (such as ion concentration) is independent of the amount of dye taken up by the cells?
2. While the authors propose concentration-induced quenching as the mechanism for the decrease in PKMDR's fluorescence lifetime at high concentrations, more experimental evidence supporting this mechanism would be valuable. For example, control experiments with structurally similar but non-quenching dyes could help confirm that the observed effect is indeed due to quenching and not other factors.
3. What is the typical imaging speed for FLIM using PKMDR in cells? Generally, TCSPC, integrated in the system TCS SP8 FALCON or STELLARIS 8 FALCON, often requires relatively long "photon counting" durations. For long-term time-lapse FLIM imaging as seen in Movie S1-3, are the lifetime values in each frame accurate, given that sufficient photon counts are essential for high goodness of fit? What are the R-squared values or Chi-squared values used for estimation of goodness of fit throughout the article?
4. Some of the data presented are based on a relatively small number of biological replicates (e.g., n=6 cells in Fig. 1,2,3. Mostly less than 10). Performing statistical tests to confirm the significance of the observed differences and including error bars or confidence intervals in the figures would strengthen the conclusions.
5. In Figure 4b, it is noticed that the TMRM-intensity and PKMDR-FLIM images do not depict 4-cell or 8-cell stages, whereas the 3D image does. Is the 3D view a representation of the fluorescence lifetime map? Additionally, the observation of fewer cells in the 2D view raises the question of whether this is due to limited depth of field, and if so, does this represent a limitation of the technique? The apparent discrepancy in the number of cells observed between the 2D and 3D views may be attributed to the limitations of depth of field in the 2D images. 2D images may only capture cells that are within a narrow focal plane, leading to an underestimate of the total cell count.
6. How was the phasor plot in Figure 5 obtained? The authors did not mention the parameter settings in the Materials and Methods section. Given that the phasor plot is a powerful tool for distinguishing between different fluorescence lifetime components, it is surprising that the authors did not utilize it to differentiate and visualize between cellular components. For the sake of clarity and reproducibility, it would be beneficial for the authors to provide a detailed description of the methodology used to generate the phasor plot, including all relevant parameter settings, and to explain why it was not employed for cellular component differentiation in this study.

7. I have noticed that in Supplementary Figures, there is a lack of spaces between some numbers and their units (e.g., “5 μ M” instead of “5 μ M”, 2.25 \pm 0.13). This inconsistency in formatting may lead to confusion or misinterpretation of the data, as proper spacing is crucial for clarity and adherence to scientific conventions. It is recommended that the authors review and revise the formatting of all numerical values and units in their supplementary materials to ensure accuracy and consistency. Such attention to detail would enhance the overall quality and readability of the research.

8. The authors should discuss the potential limitations of their method more thoroughly. For instance, what are the limitations in terms of dye loading efficiency, photostability over extended imaging sessions, and potential artifacts in complex tissue samples? Addressing these points will improve the paper’s balance and credibility.

9. As a crucial component of the research, it seems appropriate to suggest that the “Materials and Methods” section, currently located in the supplementary files, should be included within the main text of the article.

In summary, this is an interesting and well-executed study that presents a promising new tool for mitochondrial imaging. I think the manuscript has the potential to make a high impact in the field. The authors should consider the provided comments and suggestions to further strengthen their work.

Reviewer #2

(Remarks to the Author)

In this study, the authors introduced the fluorescent dye PK MitoDeepRed (PKMDR) as a tool for measuring mitochondrial membrane potential (MMP) using fluorescence lifetime imaging microscopy (FLIM). The manuscript is generally well-written, easy to follow, and presents the data clearly. However, the paper lacks significant scientific advancements and does not offer new insights into mitochondrial biology. A key concern is that while PKMDR provides an alternative method for measuring MMP, it does not demonstrate a significant advantages over existing intensity-based fluorescent probes such as TMRM, JC-1, or MitoView 633. For example, TMRM has been employed by several groups, including the O’Rourke lab (10.1074/jbc.M302673200), to monitor the dynamics of mitochondrial oscillations, demonstrating high temporal resolution. It has also shown remarkable spatial resolution, allowing the recording of mitochondrial dynamics at the single-mitochondrion level (mitochondrial flash), as demonstrated by the Wang lab (10.1016/j.cell.2008.06.017). Additionally, intensity-based fluorescent dyes like TMRM and MitoView 633 have been used to assess MMP heterogeneity in isolated cells, such as cardiomyocytes. One of the major challenges in mitochondrial biology remains the measurement of MMP *in vivo*. If the authors can provide evidence that PKMDR is capable of this, it would represent a significant advancement in the field. Some specific comments are listed below.

1. One of the advantages of FLIM is that it minimizes the effects of photon scattering in thick sample layers. However, is there a limit to the sample thickness that can be accurately measured using this technique?
2. The quantitative correlation between dye concentration and τ in live cells should be examined.
3. Figure 2, an inner membrane marker should be used in addition to the outer membrane marker TOM20.
4. The impact of temperature and pH on the behavior and dynamics of the dye should be examined, including the extent of these effects.

Reviewer #3

(Remarks to the Author)

This work introduces PKMDR, a mitochondrial dye previously developed from the corresponding author’s lab, to measure mitochondrial membrane potential (MMP) based on its concentration-dependent change of fluorescence lifetime as a result of quenching. The authors demonstrated the concentration-induced change of fluorescence lifetime of the dye in solutions and in giant unilamellar vesicles (GUV) and in response to metabolic perturbations in tissue culture cells. They further applied the dye to a variety of biological samples and argued that the dye can detect MMP changes across conditions, cell types, single cells and subcellular regions. While this work demonstrated the potential of the dye as a fluorescence-lifetime based sensor for MMP that is in principle superior than intensity based sensors, such as the widely used TMRM dye, due to fluorescence lifetime being a more robust metric against light scattering and variation of sample thickness, some of the data presented in the paper is in contradiction with the literature, calling in questions about the robustness of the probe.

Major comments:

1. The measurement of the spatial patterns of MMP in MII mouse oocytes is in contradiction with the literature (Ref [56]), which reports a lower MMP towards the cell periphery and a higher MMP towards the center measured using TMRM. This contradiction highlights potential confounding factors in the measurement of PKMDR. This paper (PMID: 31579926) has also showed that the enhanced cortical MMP is a measurement artefact. Is the dye pH sensitive? Figure 1c shows a non-monotonic behavior of the fluorescence lifetime, how to ensure the dye is operating in the quenching regime *in vivo*?
2. In general, the measurements of PKMDR should be compared with TMRM across all conditions tested in this paper to ensure robustness of the results. The authors did it for some samples but for most samples/conditions the authors compared their measurements with indirect results in the literature. For example, the authors expected MMP to increase under starvation and cited ref 33-35, which do not include any direct measurement of MMP.
3. A conceptual difficulty for this reviewer is that the authors related MMP directly with OXPHOS activity, which is not always true. For example, oligomycin increases MMP but decreases OXPHOS activity. Proton leak decreases MMP but increases respiration from FCCP treatment. The relation between OXPHOS and MMP is contextual and should not be generalized.
4. The methods used to do image analysis is not clearly stated. How are FLIM decay curves fitted: how many exponentials are used for each condition? How is mean fluorescence lifetime weighted? Is there any image segmentation before

analysis? Without this information it would be difficult to reproduce the results in the paper.

5. For a new probe, it would be useful to highlight potential limitations and usage of caution to ensure responsible and reproducible usage of the dye.

Other comments:

1. Since pH, ionic strength and other factors could vary from cell type to cell type, can the dye be used to directly compare MMP across different cell types, for example between oocyte and granulosa cells as stated in the paper?
2. Can the dye be calibrated to measure absolute value of MMP. The authors mentioned in line 290 that MMP increases linearly, which needs to be supported with calibration. This paper (PMID: 22495585) provides a calibration protocol for TMRM.
3. The metabolic heterogeneity in brain organoids and tumor slice as reported in Figure 4 should be quantified.
4. Methods section need references to the corresponding figures.
5. Since the dye binds to mitochondrial inner membrane, is the binding reversible? Does the dye measure dynamic changes of MMP, for example, does the dye diffuse out if MMP depolarizes?
6. It would be useful to show with GUV the full non-monotonic response of fluorescence lifetime to identify the quenching regime.
7. Figure S6: typo, it does not show fluorescence lifetime.

Reviewer #4

(Remarks to the Author)

Version 1:

Reviewer comments:

Reviewer #1

(Remarks to the Author)

The authors have addressed the concerns raised and provided additional data, clarifications, and methodological details that significantly strengthen the manuscript. In particular, your responses regarding the concentration dependence of PKMDR's fluorescence lifetime, the validation of quenching mechanisms, and the technical considerations for FLIM imaging (e.g., photon counts, χ^2 values, and imaging parameters) have satisfactorily resolved the initial concerns. The inclusion of error bars, additional replicates, and a more comprehensive Materials and Methods section further improves reproducibility and transparency.

The manuscript now presents a compelling and well-supported case for PKMDR-FLIM as a valuable tool for monitoring mitochondrial membrane potential and metabolic heterogeneity.

Reviewer #3

(Remarks to the Author)

The comparison between TMRM and PKMDR measurements across conditions is a significant improvement of the manuscript. The clarification of the methods and discussion of limitations also improve the manuscript. To reviewers #3 and #4, there are still outstanding questions:

1. The authors mentioned that the non-monotonic dependence of the lifetime on PKMDR concentration in Figure 1c is an artefact and provided a new Figure R2. It is unclear what is the difference of the measurement conditions between these two figures? It is also unclear if this artefact can actually impact the measurement in cells. If Fig. 1C is not representative of the real lifetime-concentration profile of PKMDR due to spectrophotometric artifacts, showing the real profile as in Fig R2, might be more useful. It is also helpful to clarify the inconsistency with Fig. 1g-j and the text lines about the monotonicity of the profile.
2. The authors addressed the question about the limitations of PKMDR, but few questions remain unanswered. What is the photostability of the dye over time (intensity and lifetime)? Since the PKMDR is washed out before imaging, does it unbind and diffuse out of mitochondria over time? ER labeling is mentioned as an unspecific effect of the dye, are there other artifacts? Given that the article studies the use of PKMDR as a membrane potential dye, it would be relevant to add in the main text the advantages/disadvantages (vs. intensity-based dyes like TMRM) and the possible limitations listed in the reply to the reviewers.
3. Due to the fact that MMP and OXPHOS activity correlation is context-dependent, the comparison of PKMDR (to measure MMP) with NADH-FLIM measurements in Ref 56 (to estimate OXPHOS activity) might need a reconsideration (specially given that TMRM measurement of MMP in Ref56 also shows the inverse result as the MMP measurements in this article from PKMDR).

4. Regarding the intercellular MMP heterogeneities in developing embryos, how are cell identities determined? In other words, how are cells numbered?
5. The authors used different linear combinations of exponential functions for the fitting (mono-, bi-, tri- exponential functions), what is the criteria or reasoning behind the specific choices? Consistency in analysis is always more welcomed. Additionally, in the introduction it should be immediately clarify that “lifetime” refers as the mean lifetime, instead of mentioning it only in the methods. Specially, given that the authors use the intensity-weighted lifetime, which large timescales have higher weight on the final mean, it is unclear if they can be readily comparable (other means, like a weighted harmonic mean, could be mathematically more robust for comparing different n-exponential). If the choice of function is not based on a chemical model of the fluorophore, one must also be aware of overfitting.
6. It would be also helpful for clarity to provide some basic equations, for example, the definition of “intensity-weighted lifetime”.
7. Is the χ^2 value provided the reduced χ^2 or the normal χ^2 ?
8. “Imaging speed” is an unclear parameter, especially given that the unit is seconds.
9. Figure R1 and Figure R4 should be added to supplementary figures.
10. According to Figure S4, PKMDR does seem to respond to changes in membrane tension, how does this impact data interpretation?
11. In Figure 5, shouldn't the distance be μm instead of nm ?

Reviewer #4

(Remarks to the Author)

Version 2:

Reviewer comments:

Reviewer #3

(Remarks to the Author)

The authors have addressed most of my questions and the paper now represents a significant contribution to the bioenergetic field. To improve the presentation of the paper, I have a few remaining questions based on the authors' response to my last review:

1. The most important point is that since the authors clarified that the increase of PKMDR lifetime with concentration in the solution is an artifact due to the set up of the fluorescence spectrometer, then why not measure the lifetime on a microscope with solution droplets, which is supposedly free of this artifact? Since Figure 1c is an important figure demonstrating the property of the dye, it will improve the paper if one includes a figure free of such artifacts. In addition, a seemingly non-monotonic curve will confuse the downstream interpretation of the data for first time readers.
2. The authors wrote in the text that TMRM suffers from non-monotonic intensity-concentration relationship, which I think is too general a statement. TMRM can operate in either quenching and non-quenching mode, whose properties have been well-characterized in the literature. I would rephrase this comment in the text in reference to the literature.
3. FLIM of NADH does not measure MMP, hence I would not use it to support the measurement of PKMDR. I would rephrase it in the text.
4. The inclusion of equations are very helpful. For eqn (2), why is the chi-squared computed in the frequency domain, shouldn't it be calculated in the time domain?

Reviewer #4

(Remarks to the Author)

Response to Reviewers:

Imaging mitochondrial membrane potential via concentration-dependent fluorescence lifetime changes

(Reference number: NCOMMS-24-81168-T)

Dilizhatai·Saimi, Luc Reymond, Tursunjan Aziz, Xuan Shen, Ziyang Luo, Shuaibo Pi, Yitong Liu, Song Fu, Shuangjin Ding, Anming Meng, Liangyi Chen, Hui Jiang, Zhixing Chen*

* E-mail: zhixingchen@pku.edu.cn

We thank the reviewers for their thoughtful and valuable comments on our original manuscript. Their feedback was extremely helpful in revising our manuscript. We have carefully considered all these comments and changed the manuscript accordingly. Please find a revised manuscript. Comments have been summarized below, along with our point-by-point response.

Reviewer #1:

Comments:

This article presents a new approach to monitor mitochondrial membrane potential (MMP) using the PKMDR probe in combination with fluorescence lifetime imaging microscopy (FLIM). The study demonstrates that PKMDR's fluorescence lifetime serves as a sensitive readout for MMP, revealing mitochondrial metabolic heterogeneity in various cell types, organoids, and tissues. The authors provide a thorough literature review on existing mitochondrial imaging tools and their limitations, setting the stage for their new method. They explain the photophysical properties of PKMDR and how its fluorescence lifetime changes with concentration. The experimental validation, involving both in vitro and in vivo systems, is sufficient, demonstrating PKMDR-FLIM's utility under various metabolic conditions and in different cell types. The findings regarding metabolic heterogeneity within and between cells, as well as the temporal dynamics of mitochondrial membrane potential, are partially novel and have good implications for mitochondrial biology and related diseases.

However, some comments and suggestions for improvement could strengthen the manuscript:

The fluorescence lifetime of a dye is generally considered to be independent of its concentration, which represents a significant advantage over fluorescence intensity measurements, as it is not affected by variations in dye concentration within cells. However, the authors present a unique probe, PKMDR, whose fluorescence lifetime specifically depends on concentration to characterize mitochondrial membrane potential.

Response:

We are grateful to the reviewer for raising this important point. Actually, PKMDR is not special – we show that the dependence of fluorescence lifetime and concentration is general among cationic dyes (Figure 1). Meanwhile, regarding the relationship between fluorescence lifetime and dye concentration, it has been reported in prior studies (Berezin & Achilefu, 2010) that fluorescence lifetime remains concentration-independent only within specific concentration ranges.

Comments:

This raises an important question: How can the authors ensure that the parameter being characterized (such as ion concentration) is independent of the amount of dye taken up by the cells?

Response:

PKMDR is a lipophilic cation, whose cellular uptake is driven by electrochemical potential. Eventually, the fluorescence lifetime is dependent of concentration, which is dependent on the dye uptake. Such uptake is ultimately dependent on the mitochondrial membrane potential.

Comments:

2. While the authors propose concentration-induced quenching as the mechanism for the decrease in PKMDR's fluorescence lifetime at high concentrations, more experimental evidence supporting this mechanism would be valuable. For example, control experiments with structurally similar but non-quenching dyes could help confirm that the observed effect is indeed due to quenching and not other factors.

Response:

We thank the reviewer for raising this question. Our experimental data (Figure 1c-g) demonstrate that the concentration-dependent decrease in fluorescence lifetime is a general property observed across various fluorophores, including TMRM, TMRE and Mitorotor-1, rather than a distinctive feature unique to PKMDR.

Comments:

3. What is the typical imaging speed for FLIM using PKMDR in cells? Generally, TCSPC, integrated in the system TCS SP8 FALCON or STELLARIS 8 FALCON, often requires relatively long "photon counting" durations. For long-term time-lapse FLIM imaging as seen in Movie S1-3, are the lifetime values in each frame accurate, given that sufficient photon counts are essential for high goodness of fit? What are the R-squared values or Chi-squared values used for estimation of goodness of fit throughout the article?

Response:

We would like to thank the reviewer for raising this question. The typical imaging speed of FLIM using PKMDR in cells depends on the parameters set at the time of imaging. When the format is 512×512 , the scan speed is 400 Hz, and the frame repetition is 10, the imaging speed is 1.498 s. For time-lapse FLIM in Movie S1-3, there is sufficient photon number in each frame for accurate fluorescence lifetime fitting. The χ^2 values of all the images in the whole article are less than 1.3.

We have made the following changes to the main text and supplementary information:

The pinhole was set to 1.0 AU. Data were collected using LAS X (Leica) at 512×512 or 1024×1024 pixel resolution and frame accumulations of 1-10 times, or collecting 500 photons per pixel. (Supplementary Table S2). FLIM data were analyzed using the LAS X software (Leica Microsystems) by fitting a mono-, bi-, or triexponential decay model (n-exponential reconvolution) to the decay ($\chi^2 < 1.3$) (Supplementary Table S2). Intensity-weighted lifetime was used as mean fluorescence lifetime in each image.

Table S2. Fluorescence microscopy data acquisition parameters.

Image	Microscope	Excitation (nm)	Objective	Scan speed (Hz)	Zoom	Pixel Size (nm)	Size (pixels)	Comment
Fig.1 e-f	SP8	633	100×1.40 oil	100	1	114	1024×1024	bi-exponential fits, 500 photons
Fig.2 c	SP8	633	100×1.40 oil	200	1	227	512×512	tri-exponential fits, 10 frame accumulation
Fig. 2 e	SP8	633	100×1.40 oil	400	1	227	512×512	tri-exponential fits, 500 photons, time-lapse interval: 15s
Fig.3 a	SP8	633	100×1.40 oil	400	1	227	512×512	tri-exponential fits, 10 frame accumulation
Fig.3 d	SP8	633	100×1.40 oil	400	1	227	512×512	tri-exponential fits, 10 frame accumulation
Fig.3 f	STELLARIS 8 FALCON	638	40x1.30 oil	100	2	142	1024×1024	tri-exponential fits, 5 frame accumulation
Fig.3 h	STELLARIS 8 FALCON	638	40x1.30 oil	100	2	142	1024×1024	tri-exponential fits, 5 frame accumulation
Fig.3 j	SP8	633	100×1.40 oil	400	1	227	512×512	tri -exponential fits, 10 frame accumulation
Fig.4 b	STELLARIS 8 FALCON	638	40x1.30 oil	100	2	142	1024×1024	tri-exponential fits, 5 frame accumulation
Fig.5 a-b	STELLARIS 8 FALCON	638	40x1.30 oil	100	2	142	1024×1024	tri-exponential fits, 5 frame accumulation
Fig.5 c	SP8	633	100×1.40 oil	400	1	227	512×512	bi-exponential fits, 10 frame accumulation
Fig.5 d-e	SP8	633	100×1.40 oil	100	1	114	1024×1024	bi-exponential fits, 500 photons
Fig.6 a	STELLARIS 8 FALCON	638	63×1.40 oil	400	1	361	512×512	tri-exponential fits, 5 frame accumulation, time-lapse interval: 15s
Fig.6 c	SP8	633	100×1.40 oil	100	1	114	1024×1024	tri-exponential fits, 500 photons, time-lapse interval: 11s

Comments:

4. Some of the data presented are based on a relatively small number of biological replicates (e.g., n=6 cells in Fig. 1,2,3. Mostly less than 10). Performing statistical tests to confirm the significance of the observed differences and including error bars or confidence intervals in the figures would strengthen the conclusions.

Response:

We have added more data and showed the error bars in the figure. We thank the reviewer for this suggestion.

We have made the following changes:

Figure 2. PKMDR fluorescence lifetime is correlated to mitochondria inner-membrane potential in live cell imaging.

d, Corresponding bar plots showing the average fluorescence lifetime change of mitochondria treated with FCCP (n = 14 cells), antimycin A (n = 14 cells), rotenone (n = 13 cells) or oligomycin (n = 13 cells). Data were presented as the mean \pm SEM. P-values were calculated using unpaired Student's t-test. ****p < 0.0001.

Figure 3. PKMDR-FLIM signal indicates the activity of mitochondrial oxidative phosphorylation.

a, PKMDR-FLIM images of HeLa cells after FBS starvation and nutrient starvation (cultured in DMEM lack of glucose, sodium-pyruvate and glutamine), and HUVECs at senescence conditions. Scale bar, 20 μ m.

b, Plots showing the average mitochondrial fluorescence lifetime under FBS starvation (n = 11 cells), nutrient starvation (n = 12 cells), and senescence HUVECs (n = 11 cells).

c, Schematic illustration of mitochondrial electron transport chain proteins.

d, PKMDR-FLIM images of wild type, COX411 and NDUFS2 knockout 143B cells.

e, Plots showing the average mitochondrial fluorescence lifetime in wild type (n = 15 cells) and knockout cells (n = 15 cells).

f, Bright field and FLIM images of PKMDR-stained mitochondria in germinal vesicle (GV) stage oocyte and granulosa cells.

g, Plots showing the average mitochondrial fluorescence lifetime in oocyte and granulosa cells (n = 8 cells).

h, Bright field, intensity and FLIM images of **PKMDR**-stained mitochondria in GV and metaphase II (MII) oocyte.

i, Plots showing the average mitochondrial fluorescence lifetime in GV and MII oocyte (n = 19 cells).

j, FLIM images of **PKMDR**-stained mitochondria in naïve and activated T cells.

k, Plots showing the average mitochondrial fluorescence lifetime in naïve and activated T cells (n = 18 cells).

Statistical analysis was performed with unpaired two-tailed Student's t-test for **b**, **e**, **i**, **k**, paired two-tailed Student's t-test for **g**. Data were presented as the mean \pm SEM. **** indicates $p < 0.0001$. Scale bars = 20 μm .

Comments:

5. In Figure 4b, it is noticed that the TMRM-intensity and PKMDR-FLIM images do not depict 4-cell or 8-cell stages, whereas the 3D image does. Is the 3D view a representation of the fluorescence lifetime map? Additionally, the observation of fewer cells in the 2D view raises the question of whether this is due to limited depth of field, and if so, does this represent a limitation of the technique? The apparent discrepancy in the number of cells observed between the 2D and 3D views may be attributed to the limitations of depth of field in the 2D images. 2D images may only capture cells that are within a narrow focal plane, leading to an underestimate of the total cell count.

Response:

We thank the reviewer for pointing this out. The 3D images represent a reconstructed fluorescence lifetime map generated from Z-stack acquisitions of 2D FLIM images. FLIM is based on confocal microscopy, which captures only a single focal plane (typically 0.5-1 μm thick). The reduced cell count observed in 2D views stems from this limitation—confocal microscopy primarily visualizes cells within a narrow focal plane, potentially missing cells above or below that plane. By performing Z-stack confocal imaging, we overcome this limitation, capturing fluorescence lifetime information across multiple axial planes to generate a comprehensive 3D representation of mitochondrial membrane potential dynamics. Simultaneous brightfield imaging during confocal acquisition was integrated to provide complementary morphological context, ensuring accurate cell counting and minimizing potential underestimation of cell numbers due to focal plane constraints.

Comments:

6. How was the phasor plot in Figure 5 obtained? The authors did not mention the parameter settings in the Materials and Methods section. Given that the phasor plot is a powerful tool for distinguishing between different fluorescence lifetime components, it is surprising that the authors did not utilize it to differentiate and visualize between cellular components. For the sake of clarity and reproducibility, it would be beneficial

for the authors to provide a detailed description of the methodology used to generate the phasor plot, including all relevant parameter settings, and to explain why it was not employed for cellular component differentiation in this study.

Response:

We are grateful to the reviewer for raising this important point. We added how the phasor plot obtained in the Methods and Material.

In our study, we employed phasor analysis to demonstrate that mitochondria within individual cells exhibit heterogeneous clustering patterns. These distinct clusters likely reflect variations in mitochondrial membrane potentials and/or metabolic states. Importantly, our observations suggest a potential spatial correlation between these functionally distinct mitochondrial subpopulations and their physical distribution within the cellular architecture. We have also made the following changes to the main text:

Phasor plot analysis in Figure 5 is performed in LAS X software (Leica Microsystems). After selecting a region of interest around the cell for analysis, the corresponding phasor plot is displayed, and pixels with similar lifetimes are grouped into clusters. Distinct clusters emerge when MMPs in an individual cell are different. Pixels forming each cluster can be identified in the lifetime image and thus separated. No thresholding is applied.

Comments:

7. I have noticed that in Supplementary Figures, there is a lack of spaces between some numbers and their units (e.g., “5 μ M” instead of “5 μ M”, 2.25 \pm 0.13). This inconsistency in formatting may lead to confusion or misinterpretation of the data, as proper spacing is crucial for clarity and adherence to scientific conventions. It is recommended that the authors review and revise the formatting of all numerical values and units in their supplementary materials to ensure accuracy and consistency. Such attention to detail would enhance the overall quality and readability of the research.

Response:

We thank the reviewer for pointing this out. We have corrected the above error.

Comments:

8. The authors should discuss the potential limitations of their method more thoroughly. For instance, what are the limitations in terms of dye loading efficiency, photostability over extended imaging sessions, and potential artifacts in complex tissue samples? Addressing these points will improve the paper’s balance and credibility.

Response:

We are grateful to the reviewer for raising this important point. We have added the following paragraph to the discussion part:

PKMDR-FLIM has certain limitations. First, the prolonged data acquisition time required for time-domain FLIM imaging, which necessitates sufficient photon accumulation for accurate lifetime calculations, inherently restricts temporal resolution. This is a particular issue at low $\Delta\Psi_m$ where PKMDR's fluorescence intensity become lower. Another limitation is that PKMDR's concentration-dependent fluorescence lifetime behavior imposes stringent experimental constraints. The probe's lifetime-concentration correlation window demand a critical threshold concentration, which can be achieved only under an optimized staining protocol outlined in our study. For non-standardized biological models, preliminary dose-response experiments are essential. While these limitations are inherent to PKMDR's design, our methodology incorporates rigorous controls and validation steps to mitigate their impact on data interpretation.

Comments:

9. As a crucial component of the research, it seems appropriate to suggest that the "Materials and Methods" section, currently located in the supplementary files, should be included within the main text of the article.

Response:

We thank the reviewer for this suggestion. We have added the materials and methods section to the main text.

Comments:

In summary, this is an interesting and well-executed study that presents a promising new tool for mitochondrial imaging. I think the manuscript has the potential to make a high impact in the field. The authors should consider the provided comments and suggestions to further strengthen their work.

Response:

Thank you for the positive assessment of this manuscript.

Reviewer #2:

In this study, the authors introduced the fluorescent dye PK MitoDeepRed (PKMDR) as a tool for measuring mitochondrial membrane potential (MMP) using fluorescence lifetime imaging microscopy (FLIM). The manuscript is generally well-written, easy to follow, and presents the data clearly. However, the paper lacks significant scientific advancements and does not offer new insights into mitochondrial biology.

Response:

We are grateful to the reviewer for raising this important point. In this study, one of our key scientific advances lies in discovering that dye concentration critically influences fluorescence lifetime, which challenges the conventional interpretation that fluorescence lifetime changes reflect mitochondrial viscosity (such as in the Mitorotor-1 probe paper (Singh et al., 2023)). This finding provides a transformative perspective in the field of biophotonics. For mitochondrial biology, on the one hand we can map the difference of mitochondrial metabolism in neurons by one-step imaging using PKMDR, which corroborates a biochemistry paper (Wei et al., 2023), showcasing that simpler imaging protocol can visualize metabolic heterogeneity in neurons. On the other hand, using PKMDR-FLIM technology, we revealed asymmetric MMP among the four blastomeres of mouse embryos at the 4-cell stage, a metabolic heterogeneity undetectable by conventional TMRM-intensity measurements. We aim to empower biologists with this tool to uncover novel mitochondrial biology insights.

Comments:

A key concern is that while PKMDR provides an alternative method for measuring MMP, it does not demonstrate a significant advantage over existing intensity-based fluorescent probes such as TMRM, JC-1, or MitoView 633. For example, TMRM has been employed by several groups, including the O'Rourke lab (10.1074/jbc.M302673200), to monitor the dynamics of mitochondrial oscillations, demonstrating high temporal resolution. It has also shown remarkable spatial resolution, allowing the recording of mitochondrial dynamics at the single-mitochondrion level (mitochondrial flash), as demonstrated by the Wang lab (10.1016/j.cell.2008.06.017). Additionally, intensity-based fluorescent dyes like TMRM and MitoView 633 have been used to assess MMP heterogeneity in isolated cells, such as cardiomyocytes.

Response:

We appreciate the reviewer's insightful comments regarding the comparative advantages of PKMDR over existing intensity-based probes like TMRM. PKMDR achieves comparable performance in monitoring mitochondrial membrane potential (MMP) dynamics with high spatiotemporal resolution, it uniquely addresses critical limitations of traditional intensity-based dyes and provides novel capabilities:

Phototoxicity artifacts: TMRM's intense phototoxicity causes fluorescence decay during prolonged imaging, confounding interpretation of MMP changes (physiological decreased vs. dye-induced damage). PKMDR's low phototoxicity eliminates this ambiguity.

Z-axis dependence: TMRM intensity shows 16% coefficient of variation across Z-positions due to optical sectioning effects.

PKMDR lifetime exhibits only 1.5% coefficient of variation (Figure R1), enabling reliable MMP quantification in 3D samples.

Thick specimen compatibility: Embryonic blastomeres at 4-/8-cell stages reside at different Z-positions. TMRM intensity measurements become unreliable due to Z-dependent signal attenuation, whereas PKMDR lifetime remains robust. Essentially this brings the measurement of MMP to another dimension.

Concentration artifacts: TMRM intensity follows a non-monotonic relationship with concentration, requiring strict pre-calibration to avoid misinterpretation (Perry, Norman, Barbieri, Brown, & Gelbard, 2011). PKMDR lifetime decreases monotonically with concentration, simplifying experimental design (Figure R2).

Figure R1. (a), FLIM (upper) and confocal (lower) image of HeLa cells stained with PKMDR and TMRM. Scale bar, 20 μ M. **(b),** Plots showing the mitochondrial fluorescence intensity and lifetime in HeLa cells at different Z position.

Figure R2. Fluorescence lifetime of PKMDR in DMSO at various concentrations showing a fluorescence lifetime decrease.

Comments:

One of the major challenges in mitochondrial biology remains the measurement of MMP in vivo. If the authors can provide evidence that PKMDR is capable of this, it would represent a significant advancement in the field. Some specific comments are listed below.

Response:

In principle PKMDR-FLIM should work in vivo. At this stage, however, we are not accessible to an upright FLIM microscope to try this experiment. We plan to try this as a future project.

Comments:

1. One of the advantages of FLIM is that it minimizes the effects of photon scattering in thick sample layers. However, is there a limit to the sample thickness that can be accurately measured using this technique?

Response:

Thank you for raising this important question. Fluorescence Lifetime Imaging Microscopy (FLIM) is inherently based on confocal microscopy, where the accurate measurement depth is closely tied to both the working distance (WD) of the objective lens and the scattering parameter of sample. Therefore, it is hard to answer this question with mathematical rigor. The working distance of Leica microscope objective (HCX PL

APO 100x/1.40 OIL) that we used in this paper is 0.13 mm. That should also be the practical distance FLIM can reach.

Comments:

2. The quantitative correlation between dye concentration and τ in live cells should be examined.

Response:

We appreciate the reviewer's insightful comment regarding the relationship between dye concentration and τ in live cells. We acknowledge that quantitatively correlating the local dye concentration with τ in this context remains challenging due to the complex microenvironment of the mitochondrial inner membrane. Current technical limitations in spatially resolving sub-mitochondrial dye concentrations at high precision further complicate this analysis. We agree that this represents an important direction for future investigation to fully unravel the biophysical mechanisms in mitochondrial studies.

Comments:

3. Figure 2, an inner membrane marker should be used in addition to the outer membrane marker TOM20.

Response:

We are grateful to the reviewer for raising this important point. We tried to co-transfect the inner mitochondrial membrane marker MCU-GFP and the outer membrane marker Tomm20-HaloTag to visualize PKMDR localization. However, we observed significant cellular stress and altered mitochondrial morphology under these conditions (fragmented and swollen mitochondria), which compromised the reliability of the co-localization analysis (Figure R3). While we were unable to achieve successful dual-labeling mitochondria, previous work has proved PKMDR's mitochondrial localization (Z. Yang et al., 2020).

Figure R3. SIM imaging of COS-7 cells expressing MCU-GFP and Tomm20-HaloTag stained with PKMDR. Scale bar, 5 μm .

Comments:

4. The impact of temperature and pH on the behavior and dynamics of the dye should be examined, including the extent of these effects.

Response:

We appreciate the reviewer's insightful suggestion. As shown in Figure R4a, the fluorescence lifetime of PKMDR was measured in phosphate buffer solutions across pH 5.8–7.8. The results demonstrated that the fluorescence lifetime remained stable and independent of environmental pH within this physiologically relevant range. To evaluate temperature effects, PKMDR was incorporated into DOPC vesicles. The fluorescence lifetime exhibited a temperature-dependent change of 0.38 ns over the range of 25°C to 45°C (Figure R4b). Importantly, this variation is significantly smaller than the 1.24 ns lifetime shift induced by concentration-dependent aggregation of PKMDR (Figure 1g). While temperature modulates the fluorescence lifetime to a measurable extent, its contribution is ~3.3-fold weaker compared to concentration-dependent effects. This indicates that PKMDR's sensing performance is primarily governed by its concentration rather than thermal fluctuations under physiological conditions.

Figure R4. Fluorescence lifetime plot of PKMDR at various pH (a) and temperature (b). Data were presented as the mean \pm SEM, n=3.

Reviewer #3 (Remarks to the Author):

This work introduces PKMDR, a mitochondrial dye previously developed from the corresponding author's lab, to measure mitochondrial membrane potential (MMP) based on its concentration-dependent change of fluorescence lifetime as a result of quenching. The authors demonstrated the concentration-induced change of fluorescence lifetime of the dye in solutions and in giant unilamellar vesicles (GUV) and in response to metabolic perturbations in tissue culture cells. They further applied the dye to a variety of biological samples and argued that the dye can detect MMP changes across conditions, cell types, single cells and subcellular regions. While this work demonstrated the potential of the dye as a fluorescence-lifetime based sensor for MMP that is in principle superior than intensity-based sensors, such as the widely used TMRM dye, due to fluorescence lifetime being a more robust metric against light scattering and variation of sample thickness, some of the data presented in the paper is in contradiction with the literature, calling in questions about the robustness of the probe.

Major comments:

1. The measurement of the spatial patterns of MMP in MII mouse oocytes is in contradiction with the literature (Ref [56]), which reports a lower MMP towards the cell periphery and a higher MMP towards the center measured using TMRM. This contradiction highlights potential confounding factors in the measurement of PKMDR. This paper (PMID: 31579926) has also showed that the enhanced cortical MMP is a measurement artefact.

Response:

We appreciate the reviewer's insightful question regarding mitochondrial membrane potential in oocytes. Our TMRM-based fluorescence measurements align closely with the findings reported in Ref56, confirming the reproducibility of MMP in oocytes (Figure R5). We acknowledge the critical concern raised in PMID:31579926 regarding potential artifacts in cortical MMP measurements using JC-1, where spatiotemporal redistribution of J-aggregates may confound interpretation. JC-1 measures MMP via ratiometric spectral shifts (green monomer vs. red J-aggregate), a process inherently susceptible to aggregation kinetics and spatial heterogeneity. In contrast, PKMDR's fluorescence lifetime decreases monotonically with increasing concentration (i.e., higher MMP), which independent of probe aggregation. This mechanistic difference ensures that PKMDR's lifetime signal is stably correlated with MMP.

On the other hand, NADH FLIM measurements (Venturas, Yang, Sakkas, & Needleman, 2023; X. Yang, Ha, & Needleman, 2021) demonstrate that the ETC flux displayed a strong spatial gradient within oocytes, with a higher flux closer to the cell periphery This finding directly corroborates the PKMDR-FLIM results

showing higher MMP (shorter lifetimes) at the oocyte periphery, suggesting a coordinated spatial gradient of mitochondrial respiration and membrane potential.

We acknowledge that further mechanistic studies are needed to dissect how MMP dynamics are physiologically regulated in oocytes and to validate these findings using complementary approaches. And we agree that current methodologies for measuring MMP in oocytes, including widely used dyes such as TMRM and JC-1 remain controversial. While our data do not resolve all methodological debates, they offer new experimental evidence and novel insights for studying spatial heterogeneity of MMP.

We have made the following changes to the main text:

Notably, the spatial distribution patterns of MMP in oocytes remain a subject of ongoing debate. Conflicting results have been reported between TMRM measurements (showing MMP gradient decreasing from the central region to the periphery) and those obtained through JC-1 imaging and NADH FLIM (demonstrating an inverse peripheral-to-central gradient)⁵⁴⁻⁵⁷. Our analysis aligns closely with the spatial patterns observed in JC-1 and NADH FLIM studies, which providing support and new insights for the study of spatial heterogeneity of MMP.

Figure R5. FLIM and confocal image of oocyte co-stained with PKMDR and TMRM. Scale bar, 20 μ M.

Comments:

Is the dye pH sensitive?

Response:

We measured the fluorescence lifetime of PKMDR in phosphate buffer solutions across pH 5.8–7.8. As shown in Figure R4a, the fluorescence lifetime of PKMDR is not affected by pH.

Figure R4. Fluorescence lifetime plot of PKMDR at various pH (a) and temperature (b). Data were presented as the mean \pm SEM, $n=3$.

Comments:

Figure 1c shows a non-monotonic behavior of the fluorescence lifetime, how to ensure the dye is operating in the quenching regime in vivo?

Response:

Regarding the concern about the monotonic decrease in PKMDR fluorescence lifetime, we acknowledge that Figure 1c might initially appear misleading due to artifacts introduced by the spectrophotometric measurements. Specifically, radiative energy transport—a phenomenon where re-absorption of emitted fluorescence photons by neighboring dye molecules distorts lifetime measurements (Kelley & Kelley, 2022; Langhals & Schlücker, 2022)—can artificially inflate fluorescence lifetimes as the concentrations increased, creating an upward trend. However, this artifact does not reflect the true relationship between concentration and fluorescence lifetime. As demonstrated in Figure R2, the correlation between PKMDR concentration and fluorescence lifetime is actually monotonically decreasing. The vesicle experiments in Figures 1g-j provide direct evidence of this relationship. These results confirm that fluorescence lifetime remains constant below the critical concentration, and only decreases monotonically once this critical concentration is exceeded. Notably, PKMDR exhibits a significantly lower critical concentration compared to conventional dyes like TMRM or TMRE, owing to its enhanced lipophilicity. This property ensures that at the recommended staining concentrations (optimized in our study), PKMDR’s fluorescence lifetime reliably reflects membrane potential changes in live cells, with no artificial artifacts arising from radiative energy transport.

Figure R2. Fluorescence lifetime of PKMDR in DMSO at various concentrations showing a fluorescence lifetime decrease.

Comments:

2. In general, the measurements of PKMDR should be compared with TMRM across all conditions tested in this paper to ensure robustness of the results. The authors did it for some samples but for most samples/conditions the authors compared their measurements with indirect results in the literature. For example, the authors expected MMP to increase under starvation and cited ref 33-35, which do not include any direct measurement of MMP.

Response:

We thank the reviewer for this suggestion. We have added the experimental results of TMRM to detect the changes of MMP after adding FCCP, antimycin a, rotenone, oligomycin or starvation treatment or induction of senescence. We also used TMRM to detect the MMP difference mitochondria of 143 B cells between normal and knockout NDUFS2 and COX4I1. We have made the following changes to the supplementary information:

Figure S3. Measuring membrane potential using TMRM.

(a), Confocal image of HeLa cells treated with FCCP (OXPHOS uncouplers), antimycin A (complex III inhibitor), rotenone (complex I inhibitor) and oligomycin (ATP synthase inhibitor). Scale bar, 20 μ M. (b), Corresponding bar plots showing the average fluorescence intensity change of mitochondria treated with FCCP (n = 9 cells), antimycin A (n = 11 cells), rotenone (n = 9 cells) or oligomycin (n = 14 cells). Data were presented as the mean \pm SEM. P-values were calculated using unpaired Student's t-test. ****p < 0.0001.

Figure S6. Mitochondrial membrane potential in starvation and senescence cell.

(a) TMRM-Intensity images of HeLa cells after FBS starvation and nutrient starvation (cultured in DMEM lack of glucose, sodium-pyruvate and glutamine), and HUVECs at senescence conditions. Scale bar, 20 μ m. **(b)** Plots showing the average mitochondrial fluorescence intensity under FBS starvation (n = 15 cells), nutrient starvation (n = 17 cells), and senescence HUVECs (n = 20 cells).

Figure S7. Mitochondrial membrane potential in COX4I1 and NDUF52 knockout 143B cells.

(a) TMRM-intensity images of wild type, COX4I1 and NDUF52 knockout 143B cells. Scale bar, 20 μ m. **(b)** Plots showing the average mitochondrial fluorescence lifetime in wild type (n = 20 cells) and knockout cells (n = 22 cells). P-values were calculated using unpaired Student's t-test. ****p < 0.0001.

Comments:

3. A conceptual difficulty for this reviewer is that the authors related MMP directly with OXPHOS activity, which is not always true. For example, oligomycin increases MMP but decreases OXPHOS activity. Proton leak decreases MMP but increases respiration from FCCP treatment. The relation between OXPHOS and MMP is contextual and should not be generalized.

Response:

Thank you for raising this important conceptual concern. In Figure 2, we explicitly focused on measuring direct changes in MMP induced by mitochondrial drugs (e.g., FCCP, antimycin A, rotenone, oligomycin). All descriptions in this section strictly refer to "increased/decreased MMP," as these compounds directly alter proton gradient dynamics (e.g., FCCP dissipates MMP, Oligomycin elevates MMP by inhibiting ATP synthase).

In Figure 3, however, the models (e.g., serum/nutrient starvation, senescence, and knockdown of electron transport chain proteins) were selected based on prior literature demonstrating their systemic impact on oxidative phosphorylation (OXPHOS) activity. For example, knocking down the electron transport chain proteins is known to downregulate OXPHOS capacity. Here, we used the term "OXPHOS activity" to reflect the broader functional consequences of these interventions.

We fully agree that MMP and OXPHOS activity are context-dependent and not universally correlated. For example, as noted, oligomycin elevates MMP while suppressing OXPHOS, and proton leak uncouples MMP from respiration. Thus, we revised the expression in lines 258, 259, 289. Thank you again for highlighting this critical nuance, which strengthens the interpretation of our findings.

Comments:

4. The methods used to do image analysis is not clearly stated. How are FLIM decay curves fitted: how many exponentials are used for each condition? How is mean fluorescence lifetime weighted? Is there any image segmentation before analysis? Without this information it would be difficult to reproduce the results in the paper.

Response:

We would like to thank the reviewer for raising this question. We analyzed the FLIM image using the LAS X software (Leica Microsystems) by fitting a bi-, or triexponential decay model (n-exponential reconvolution) to the decay ($\chi^2 < 1.3$) (Supplementary Table S2). No image segmentation processing was performed. Some FLIM images selected ROI for analysis. Intensity-weighted lifetime was used as mean fluorescence lifetime in each image.

We have made the following changes to the main text and supplementary information:

The pinhole was set to 1.0 AU. Data were collected using LAS X (Leica) at 512×512 or 1024×1024 pixel resolution and frame accumulations of 1-10 times, or collecting 500 photons per pixel. (Supplementary Table S2). FLIM data were analyzed using the LAS X software (Leica Microsystems) by fitting a mono-, bi-, or triexponential decay model (n-exponential reconvolution) to the decay ($\chi^2 < 1.3$) (Supplementary Table S2). Intensity-weighted lifetime was used as mean fluorescence lifetime in each image.

Table S2. Fluorescence microscopy data acquisition parameters.

Image	Microscope	Excitation (nm)	Objective	Scan speed (Hz)	Zoom	Pixel Size (nm)	Size (pixels)	Comment
Fig.1 e-f	SP8	633	100×1.40 oil	100	1	114	1024×1024	bi-exponential fits, 500 photons
Fig.2 c	SP8	633	100×1.40 oil	200	1	227	512×512	tri-exponential fits, 10 frame accumulation
Fig. 2 e	SP8	633	100×1.40 oil	400	1	227	512×512	tri-exponential fits, 500 photons, time-lapse interval: 15s
Fig.3 a	SP8	633	100×1.40 oil	400	1	227	512×512	tri-exponential fits, 10 frame accumulation
Fig.3 d	SP8	633	100×1.40 oil	400	1	227	512×512	tri-exponential fits, 10 frame accumulation
Fig.3 f	STELLARIS 8 FALCON	638	40x1.30 oil	100	2	142	1024×1024	tri-exponential fits, 5 frame accumulation
Fig.3 h	STELLARIS 8 FALCON	638	40x1.30 oil	100	2	142	1024×1024	tri-exponential fits, 5 frame accumulation
Fig.3 j	SP8	633	100×1.40 oil	400	1	227	512×512	tri -exponential fits, 10 frame accumulation
Fig.4 b	STELLARIS 8 FALCON	638	40x1.30 oil	100	2	142	1024×1024	tri-exponential fits, 5 frame accumulation
Fig.5 a-b	STELLARIS 8 FALCON	638	40x1.30 oil	100	2	142	1024×1024	tri-exponential fits, 5 frame accumulation
Fig.5 c	SP8	633	100×1.40 oil	400	1	227	512×512	bi-exponential fits, 10 frame accumulation
Fig.5 d-e	SP8	633	100×1.40 oil	100	1	114	1024×1024	bi-exponential fits, 500 photons
Fig.6 a	STELLARIS 8 FALCON	638	63×1.40 oil	400	1	361	512×512	tri-exponential fits, 5 frame accumulation, time-lapse interval: 15s
Fig.6 c	SP8	633	100×1.40 oil	100	1	114	1024×1024	tri-exponential fits, 500 photons, time-lapse interval: 11s

Comments:

5. For a new probe, it would be useful to highlight potential limitations and usage of caution to ensure responsible and reproducible usage of the dye.

Response:

We thank the reviewer for pointing this out. We have added the following paragraph to the discussion part:

PKMDR-FLIM exhibits certain limitations. First, the prolonged data acquisition time required for time-domain FLIM imaging, which necessitates sufficient photon accumulation for accurate lifetime calculations, inherently restricts temporal resolution. At low $\Delta\Psi_m$, PKMDR's fluorescence intensity decreases significantly, necessitating repeated scans to achieve adequate signal-to-noise ratios—a trade-off that further compromises temporal resolution. Another limitation is that PKMDR's concentration-dependent fluorescence lifetime behavior imposes stringent experimental constraints. The probe's lifetime-concentration correlation becomes reliable only above a critical threshold concentration, requiring users to strictly adhere to the optimized staining protocol outlined in our study. For non-standardized biological models, preliminary dose-response experiments are essential. While these limitations are inherent to PKMDR's design, our methodology incorporates rigorous controls and validation steps to mitigate their impact on data interpretation.

In the Material and Methods section, we explicitly emphasized the protocols and usage of caution for PKMDR:

PKMDR labeling for live-cell FLIM imaging

Cells were seeded into a glass-bottom dish (Cellvis) a day before imaging with the cell density at around 4×10^4 cells/mL. For labeling of mitochondria, cells were stained with DMEM supplemented with 250 nM PKMDR for 20 min at 37 °C. Following the staining procedure, the cells were washed three times with pre-warmed PBS or culture medium to remove the unbound dye. If long-term time-lapse imaging is desired, labeling density should be kept as low as possible for optimal imaging. If the signal is too weak, leading to noisy data or low intramitochondrial contrast, we recommend extending the staining time up to 45 min. Moreover, we observed that a washing step of 15 to 30 min following the staining procedure could significantly increase the intramitochondrial contrast while reducing unspecific ER labeling. For multi-color labeling, we recommend sequential staining if the used dyes require different incubation times.

Other comments:

1. Since pH, ionic strength and other factors could vary from cell type to cell type, can the dye be used to directly compare MMP across different cell types, for example between oocyte and granulosa cells as stated in the paper?

Response:

We are grateful to the reviewer for raising this important point. In our study, we tested PKMDR fluorescence lifetime across a range of in vitro pH gradients (pH 5.8–7.8) (Figure R4a), confirming that its lifetime response remained stable and correlated with MMP changes induced by FCCP, antimycin A or oligomycin (see Figure 2). Thus, PKMDR can compare MMP between cell types (e.g., oocytes vs. granulosa cells) provided experimental conditions are strictly standardized.

Comments:

2. Can the dye be calibrated to measure absolute value of MMP. The authors mentioned in line 290 that MMP increases linearly, which needs to be supported with calibration. This paper (PMID: 22495585) provides a calibration protocol for TMRM.

Response:

We would like to thank the reviewer for raising this question. Currently, due to technical limitations, we are unable to calibrate the dye to report absolute values of MMP at this stage. This aspect represents an important direction for future research and requires further optimization and validation with established reference methods, which are beyond the scope of the current study. Concerning the statement on line 290, we have made the following changes to the main text:

In these spreading cells, mitochondrial membrane potential increased from the nuclear region to the cell periphery.

Comments:

3. The metabolic heterogeneity in brain organoids and tumor slice as reported in Figure 4 should be quantified.

Response:

The primary objective of this figure is to demonstrate that our PKMDR probe can effectively visualize MMP not only in cultured cells but also in complex biological systems like brain organoids and tumor tissue slices. While the current data qualitatively reveal the inherent heterogeneity of MMP distribution within these models, we fully acknowledge the importance of quantitative characterization. Unfortunately, standardized methodologies for spatial quantification of metabolic heterogeneity in organoids and thick tissue specimens remain underdeveloped in the field. Addressing this technical challenge will require systematic optimization of imaging analysis pipelines and heterogeneity metrics, which we plan to pursue in future studies specifically focused on computational bioenergetics profiling.

Comments:

4. Methods section need references to the corresponding figures.

Response:

We thank the reviewer for this suggestion. We have corrected the above error.

Comments:

5. Since the dye binds to mitochondrial inner membrane, is the binding reversible? Does the dye measure dynamic changes of MMP, for example, does the dye diffuse out if MMP depolarizes?

Response:

Thank you for raising this important question regarding the reversibility and dynamic response of PKMDR. As a lipophilic cation, PKMDR's localization to the mitochondrial inner membrane is entirely driven by MMP, and its binding is fully reversible. This behavior aligns with the Nernstian distribution mechanism typical of cationic probes, where accumulation and release are governed by changes in MMP. In Figure 6a, long-term live-cell imaging reveals temporal changes in PKMDR fluorescence lifetime, reflecting dynamic fluctuations in MMP over time. These observations confirm that PKMDR responds reversibly to physiological variations in MMP. In Figure 2c, PKMDR rapidly dissociates from the mitochondrial inner membrane and diffuses into the cytosol after treated with FCCP. This confirms that PKMDR's localization is directly tied to MMP, and PKMDR will diffuse out if MMP depolarizes.

Comments:

6. It would be useful to show with GUV the full non-monotonic response of fluorescence lifetime to identify the quenching regime.

Response:

We are grateful to the reviewer for raising this important point. Regarding the concern about the monotonic decrease in PKMDR fluorescence lifetime, we acknowledge that Figure 1c might initially appear misleading due to artifacts introduced by the spectrophotometric measurements. Specifically, radiative energy transport—a phenomenon where re-absorption of emitted fluorescence photons by neighboring dye molecules distorts lifetime measurements (Kelley & Kelley, 2022; Langhals & Schlücker, 2022)—can artificially inflate fluorescence lifetimes as the concentrations increased, creating an upward trend. However, this artifact does not reflect the actual relationship between concentration and fluorescence lifetime.

As demonstrated in Figure R2, the actual correlation between PKMDR concentration and fluorescence lifetime is indeed monotonically decreasing. The vesicle experiments in Figures 1g-j provide direct evidence of this relationship. These results confirm that fluorescence lifetime remains constant below the critical concentration, and only decreases monotonically once this critical concentration is exceeded. Importantly, PKMDR exhibits a significantly lower critical concentration compared to conventional dyes like TMRM or TMRE, owing to its enhanced lipophilicity. This property ensures that at the recommended staining concentrations (optimized in our study), PKMDR's fluorescence lifetime reliably reflects membrane potential changes in live cells, with no artificial artifacts arising from radiative energy transport.

Figure R2. Fluorescence lifetime of PKMDR in DMSO at various concentrations showing a fluorescence lifetime decrease.

Comments:

7. Figure S6: typo, it does not show fluorescence lifetime.

Response:

We thank the reviewer for pointing this out. We have corrected the above error.

Reviewer #4 (Remarks to the Author):

- Berezin, M. Y., & Achilefu, S. (2010). Fluorescence Lifetime Measurements and Biological Imaging. *Chemical Reviews*, *110*(5), 2641-2684. doi:10.1021/cr900343z
- Kelley, A. M., & Kelley, D. F. (2022). Comment on “Dependence of the Fluorescent Lifetime τ on the Concentration at High Dilution”. *The Journal of Physical Chemistry Letters*, *13*(51), 11942-11945. doi:10.1021/acs.jpcclett.2c02677
- Langhals, H., & Schlücker, T. (2022). Dependence of the Fluorescent Lifetime τ on the Concentration at High Dilution. *The Journal of Physical Chemistry Letters*, *13*(32), 7568-7573. doi:10.1021/acs.jpcclett.2c01447
- Perry, S. W., Norman, J. P., Barbieri, J., Brown, E. B., & Gelbard, H. A. (2011). Mitochondrial membrane potential probes and the proton gradient: a practical usage guide. *BioTechniques*, *50*(2), 98-115. doi:10.2144/000113610
- Singh, G., George, G., Raja, S. O., Kandaswamy, P., Kumar, M., Thutupalli, S., . . . Gulyani, A. (2023). A molecular rotor FLIM probe reveals dynamic coupling between mitochondrial inner membrane fluidity and cellular respiration. *Proc Natl Acad Sci U S A*, *120*(24), e2213241120. doi:10.1073/pnas.2213241120
- Venturas, M., Yang, X., Sakkas, D., & Needleman, D. (2023). Noninvasive metabolic profiling of cumulus cells, oocytes, and embryos via fluorescence lifetime imaging microscopy: a mini-review. *Human Reproduction*, *38*(5), 799-810. doi:10.1093/humrep/dead063
- Wei, Y., Miao, Q., Zhang, Q., Mao, S., Li, M., Xu, X., . . . Hu, G. (2023). Aerobic glycolysis is the predominant means of glucose metabolism in neuronal somata, which protects against oxidative damage. *Nature Neuroscience*, *26*(12), 2081-2089. doi:10.1038/s41593-023-01476-4
- Yang, X., Ha, G., & Needleman, D. J. (2021). A coarse-grained NADH redox model enables inference of subcellular metabolic fluxes from fluorescence lifetime imaging. *eLife*, *10*. doi:10.7554/eLife.73808
- Yang, Z., Li, L., Ling, J., Liu, T., Huang, X., Ying, Y., . . . Chen, Z. (2020). Cyclooctatetraene-conjugated cyanine mitochondrial probes minimize phototoxicity in fluorescence and nanoscopic imaging. *Chem Sci*, *11*(32), 8506-8516. doi:10.1039/d0sc02837a

Response to Reviewers:

Imaging mitochondrial membrane potential via concentration-dependent fluorescence lifetime changes

(Reference number: NCOMMS-24-81168A)

Dilizhatai·Saimi, Luc Reymond, Tursunjan Aziz, Xuan Shen, Ziyang Luo, Shuaibo Pi, Yitong Liu, Song Fu, Shuangjin Ding, Anming Meng, Liangyi Chen, Hui Jiang, Zhixing Chen^{*}

* E-mail: zhixingchen@pku.edu.cn

Thank you very much for your time, insightful comments, and constructive feedback. We are deeply grateful for the editor's and reviewers' thoughtful evaluation of our manuscript (NCOMMS-24-81168A). We have carefully considered all these comments and changed the manuscript accordingly. Please find a revised manuscript. Comments have been summarized below, along with our point-by-point response.

REVIEWER COMMENTS

Reviewer #1 (Remarks to the Author):

Comment:

The authors have addressed the concerns raised and provided additional data, clarifications, and methodological details that significantly strengthen the manuscript. In particular, your responses regarding the concentration dependence of PKMDR's fluorescence lifetime, the validation of quenching mechanisms, and the technical considerations for FLIM imaging (e.g., photon counts, χ^2 values, and imaging parameters) have satisfactorily resolved the initial concerns. The inclusion of error bars, additional replicates, and a more comprehensive Materials and Methods section further improves reproducibility and transparency.

The manuscript now presents a compelling and well-supported case for PKMDR-FLIM as a valuable tool for monitoring mitochondrial membrane potential and metabolic heterogeneity.

Response:

Thank you for the positive assessment of this manuscript.

Reviewer #3 (Remarks to the Author):

Comments:

The comparison between TMRM and PKMDR measurements across conditions is a significant improvement of the manuscript. The clarification of the methods and discussion of limitations also improve the manuscript. To reviewers #3 and #4, there are still outstanding questions:

1. The authors mentioned that the non-monotonic dependence of the lifetime on PKMDR concentration in Figure 1c is an artefact and provided a new Figure R2. It is unclear what is the difference of the measurement conditions between these two figures? It is also unclear if this artefact can actually impact the measurement in cells. If Fig. 1C is not representative of the real lifetime-concentration profile of PKMDR due to

spectrophotometric artifacts, showing the real profile as in Fig R2, might be more useful. It is also helpful to clarify the inconsistency with Fig. 1g-j and the text lines about the monotonicity of the profile.

Response:

We apologize for the confusion. The artifact is clearly explained in Reference 29. This artifact only happened when the experiments were performed in the common 90° geometry (Figure RR1a), where the excitation beam enters the cell along one axis and the fluorescence are collected from roughly the middle of the cell, propagating along a perpendicular axis. When using this device to detect fluorescence lifetime, the absorption and re-emission of the fluorescence (“radiative energy transport”) lengthens the apparent lifetime at higher concentrations. Therefore, when measuring cells and GUVs (Figure 1g-j), due to the shorter and difference of light path, this artifact will not exist on the fluorescence lifetime imaging microscope (Figure RR1b). Figure R2 is intended to illustrate what the data in Figure 1c should look like in the absence of artifacts. In other words, Figure R2 was only illustrative and thus should not be added in the manuscript.

Figure RR1. The light path of fluorescence spectrometer (a) and fluorescence lifetime imaging microscope (b).

Comment:

2. The authors addressed the question about the limitations of PKMDR, but few questions remain unanswered. What is the photostability of the dye over time (intensity and lifetime)? Since the PKMDR is washed out before imaging, does it unbind and diffuse out of mitochondria over time? ER labeling is mentioned as an unspecific effect of the dye, are there other artifacts? Given that the article studies the use of PKMDR as

a membrane potential dye, it would be relevant to add in the main text the advantages/disadvantages (vs. intensity-based dyes like TMRM) and the possible limitations listed in the reply to the reviewers.

Response:

We thank the reviewer for raising this question. The photostability of PKMDR has been discussed in Reference 23. The localization of PKMDR to mitochondria is dependent on the MMP. Unless the MMP is altered, PKMDR will not diffuse out of mitochondria after washing. Actually, when using PKMDR for staining, ER labeling hardly occurs, and other artifacts do not appear either. In the previous review comments, we were required to highlight potential limitations and usage precautions to ensure the responsible and reproducible application of the dye. Therefore, referring to other mitochondrial probes(Liu et al., 2022), we wrote about the possible issues and corresponding solutions.

We have made the following changes to the main text and supplementary information:

Compared to the intensity signal of TMRM, PKMDR's low phototoxicity eliminates fluorescence decay artifacts during prolonged imaging. Moreover, PKMDR lifetime exhibits minimal Z position variation (1.5% coefficient of variation vs. TMRM intensity's 16% coefficient of variation), enabling reliable 3D quantification in specimens like embryonic blastomeres at different depths where TMRM becomes unreliable (Figure S11). Furthermore, PKMDR lifetime decreases monotonically with concentration, simplifying experimental design and avoiding the misinterpretation risks inherent in TMRM's non-monotonic intensity-concentration relationship.

Figure S11. Measurement of mitochondrial membrane potential using TMRM-intensity and PKMDR-FLIM. **(a)** FLIM (upper) and confocal (lower) image of HeLa cells stained with PKMDR and TMRM. Scale bar, 20 μ M. **(b)** Plots showing the mitochondrial fluorescence intensity and lifetime in HeLa cells at different Z position.

Comments:

3. Due to the fact that MMP and OXPHOS activity correlation is context-dependent, the comparison of PKMDR (to measure MMP) with NADH-FLIM measurements in Ref 56 (to estimate OXPHOS activity) might need a reconsideration (specially given that TMRM measurement of MMP in Ref56 also shows the inverse result as the MMP measurements in this article from PKMDR).

Response:

We are grateful to the reviewer for raising this important point. Our PKMDR-FLIM data revealing mitochondrial heterogeneity in oocytes suggest a potential gradient distribution of mitochondrial membrane potential from the peripheral to central regions of oocytes. We agree that this could be controversial, and this phenomenon warrants more in-depth investigation to fully elucidate its underlying mechanisms. We see this data as a new clues and would like to present it to researchers in the oocyte biology field to stimulate further studies.

Comments:

4. Regarding the intercellular MMP heterogeneities in developing embryos, how are cell identities determined? In other words, how are cells numbered?

Response:

We thank the reviewer for pointing this out. During the 2-cell, 4-cell, and 8-cell stages of an embryo, the number of cells can be clearly observed under a microscope, allowing for manual numbering.

Comments:

5. The authors used different linear combinations of exponential functions for the fitting (mono-, bi-, tri- exponential functions), what is the criteria or reasoning behind the specific choices? Consistency in analysis is always more welcomed. Additionally, in the introduction it should be immediately clarify that “lifetime” refers as the mean lifetime, instead of mentioning it only in the methods. Specially, given that the authors use the intensity-weighted lifetime, which large timescales have higher weight on the final mean, it is unclear if they can be readily comparable (other means, like a weighted harmonic mean, could be mathematically more robust for comparing different n-exponential). If the choice of function is not based on a chemical model of the fluorophore, one must also be aware of overfitting.

Response:

We appreciate the reviewer’s insightful question regarding fluorescence lifetime fitting. For the FLIM analysis, we observed that different samples required distinct exponential functions to achieve optimal goodness-of-fit. Specifically, some FLIM data were fitted with biexponential decay model to achieve χ^2 value below our acceptance criterion of <1.3 , whereas others necessitated tri-exponential functions. This approach aligns with established practices in FLIM research, as demonstrated in previous studies (Frei, Koch, Hiblot, & Johnsson, 2022; M. S. Frei et al., 2022; Singh et al., 2023). Furthermore, all lifetime values reported in our manuscript, both in images and tables, represent the intensity-weighted average fluorescence lifetimes. This is the conventional and widely accepted metric for reporting and comparing lifetimes in biological FLIM imaging. To ensure experimental consistency across comparative analyses, we maintained a uniform fitting approach for all relevant figures, thereby preserving the reliability and comparability of our results.

Comments:

6. It would be also helpful for clarity to provide some basic equations, for example, the definition of “intensity-weighted lifetime”.

Response:

We appreciate the reviewer's insightful suggestion. We have made the following changes to the main text:

FLIM data were analyzed using the LAS X software (Leica Microsystems) by fitting a mono-, bi-, or triexponential decay model (n-exponential reconvolution, equation 1) to the decay ($\chi^2 < 1.3$, equation 2) (Supplementary Table S2). Intensity-weighted lifetime (equation 3) was used as mean fluorescence lifetime in each image.

$$y(t) = \{IRF(t + Shift_{IRF}) + Bkgr_{IRF}\} \otimes \left\{ \sum_{i=1}^{n-1} A[i] e^{-\frac{t}{\tau[i]}} + Bkgr \right\} \quad (1)$$

n : Number of exponential components

A : Amplitudes – Exponential pre- factors

τ : Exponential Decay Times (e.g. lifetimes)

Bkgr : Tail Offset – Correction for background

Shift_{IRF} : IRF Shift – Correction for IRF displacement

Bkgr_{IRF} : Irf Background – Correction for IRF background

$$\chi^2 = \frac{1}{\nu} \left\{ \sum_{j=1}^N \left[\frac{\varphi^j(\omega) - \varphi^C(\omega)}{\sigma_\phi} \right]^2 + \sum_{j=1}^N \left[\frac{m^j(\omega) - m^C(\omega)}{\sigma_M} \right]^2 \right\} \quad (2)$$

N: Total number of frequencies

ν : Number of degrees of freedom of the system. Since the number of data points is twice the number of frequencies, $\nu = 2N - p$ where p is the number of variables.

σ_ϕ , σ_M : Uncertainties used in the phase and modulation values. It was found that the experimental result is not strongly dependent on σ_ϕ and σ_M .

$\varphi^j(\omega)$: Measured frequency-dependent values of phase angle.

$m^j(\omega)$: Measured frequency-dependent values of demodulation.

$\varphi^C(\omega)$: Calculated frequency-dependent values of phase angle.

$m^C(\omega)$: Calculated frequency-dependent values of demodulation.

$$\tau_{Av Int} = \frac{\sum_{k=0}^{n-1} I[k] \tau[k]}{I_{sum}} \quad (3)$$

I : Intensities associated with each exponential component, normalized to the time resolution of the measured decay curve in order to be displayed in photon counts

I_{Sum} : of Fluorescence Intensity for all components

$\tau_{Av Int}$: Mean Photon Arrival Time – Intensity weighted average lifetime

Comments:

7. Is the χ^2 value provided the reduced χ^2 or the normal χ^2 ?

Response:

We are grateful to the reviewer for pointing this out. The χ^2 value provided in the manuscript is reduced χ^2 .

Comments:

8. “Imaging speed” is an unclear parameter, especially given that the unit is seconds.

Response:

We thank the reviewer for pointing this out. There was a writing error in the last response letter, which should be:

When the format is 512×512 , the scan speed is 400 Hz, and the frame repetition is 10, the imaging speed is 1.498 s/frame.

Comments:

9. Figure R1 and Figure R4 should be added to supplementary figures.

Response:

We are grateful to the reviewer for pointing this out. We have made the following changes to the main text and supplementary information:

Moreover, we also examined the impact of temperature and pH on the lifetime of **PKMDR**. The fluorescence lifetime of **PKMDR** was measured in phosphate buffer solutions across pH 5.8–7.8. The results demonstrated that the fluorescence lifetime remained stable and independent of environmental pH within this physiologically relevant range (Supplementary Fig. S3a). To evaluate temperature effects, **PKMDR** was incorporated into DOPC vesicles. The fluorescence lifetime exhibited a temperature-dependent change of 0.38 ns over the range of 25°C to 45°C

(Supplementary Fig. S3b). Importantly, this variation is significantly smaller than the 1.24 ns lifetime shift induced by concentration-dependent aggregation of **PKMDR** (Figure 1g). While temperature modulates the fluorescence lifetime to a measurable extent, its contribution is ~3.3-fold weaker compared to concentration-dependent effects. This indicates that **PKMDR**'s sensing performance is primarily governed by its concentration rather than thermal fluctuations under physiological conditions.

Figure S3. Fluorescence lifetime plot of PKMDR at various pH (a) and temperature (b). Data were presented as the mean \pm SEM, n=3

Compared to the intensity signal of TMRM, PKMDR's low phototoxicity eliminates fluorescence decay artifacts during prolonged imaging. Moreover, PKMDR lifetime exhibits minimal Z position variation (1.5% coefficient of variation vs. TMRM intensity's 16% coefficient of variation), enabling reliable 3D quantification in specimens like embryonic blastomeres at different depths where TMRM becomes unreliable (Figure S11). Furthermore, PKMDR lifetime decreases monotonically with concentration, simplifying experimental design and avoiding the misinterpretation risks inherent in TMRM's non-monotonic intensity-concentration relationship.

Figure S11. Measurement of mitochondrial membrane potential using TMRM-intensity and PKMDR-FLIM. (a) FLIM (upper) and confocal (lower) image of HeLa cells stained with PKMDR and TMRM. Scale bar, 20 μ M. (b) Plots showing the mitochondrial fluorescence intensity and lifetime in HeLa cells at different Z position.

Comments:

10. According to Figure S4, PKMDR does seem to respond to changes in membrane tension, how does this impact data interpretation?

Response:

We are grateful to the reviewer for raising this important point. In Figure S4, we added hypertonic and hypotonic solutions to decrease and increase membrane tension, respectively. If PKMDR responded to membrane tension, it should have induced two distinct changes in fluorescence lifetime. However, our experimental results showed that fluorescence lifetime increased in both hypertonic and hypotonic solutions. We attribute this to the deterioration of cellular physiological state and reduction in membrane potential caused by adding solutions with different osmotic pressures, rather than a response to membrane tension.

Comments:

11. In Figure 5, shouldn't the distance be μ m instead of nm?

Response:

We thank the reviewer for pointing this out. We have corrected the above error.

Reviewer #4 (Remarks to the Author):

- Frei, M. S., Koch, B., Hiblot, J., & Johnsson, K. (2022). Live-Cell Fluorescence Lifetime Multiplexing Using Synthetic Fluorescent Probes. *ACS Chemical Biology*, *17*(6), 1321-1327. doi:10.1021/acscchembio.2c00041
- Frei, M. S., Tarnawski, M., Roberti, M. J., Koch, B., Hiblot, J., & Johnsson, K. (2022). Engineered HaloTag variants for fluorescence lifetime multiplexing. *Nat Methods*, *19*(1), 65-70. doi:10.1038/s41592-021-01341-x
- Liu, T., Stephan, T., Chen, P., Keller-Findeisen, J., Chen, J., Riedel, D., . . . Chen, Z. (2022). Multi-color live-cell STED nanoscopy of mitochondria with a gentle inner membrane stain. *119*(52), e2215799119. doi:doi:10.1073/pnas.2215799119
- Singh, G., George, G., Raja, S. O., Kandaswamy, P., Kumar, M., Thutupalli, S., . . . Gulyani, A. (2023). A molecular rotor FLIM probe reveals dynamic coupling between mitochondrial inner membrane fluidity and cellular respiration. *Proc Natl Acad Sci U S A*, *120*(24), e2213241120. doi:10.1073/pnas.2213241120

Response to Reviewers:

Imaging mitochondrial membrane potential via concentration-dependent fluorescence lifetime changes

(Reference number: NCOMMS-24-81168B)

Dilizhatai·Saimi, Luc Reymond, Tursunjan Aziz, Xuan Shen, Ziyang Luo, Shuaibo Pi, Yitong Liu, Song Fu, Shuangjin Ding, Anming Meng, Liangyi Chen, Hui Jiang, Zhixing Chen*

* E-mail: zhixingchen@pku.edu.cn

Thank you very much for your time, insightful comments, and constructive feedback. We are deeply grateful for the editor's and reviewers' thoughtful evaluation of our manuscript (NCOMMS-24-81168B). We have carefully considered all these comments and changed the manuscript accordingly. Please find a revised manuscript. Comments have been summarized below, along with our point-by-point response.

Remarks to Authors from Referee #3 on the NCOMMS-24-81168B version

Comment:

The authors have addressed most of my questions and the paper now represents a significant contribution to the bioenergetic field. To improve the presentation of the paper, I have a few remaining questions based on the authors' response to my last review:

1. The most important point is that since the authors clarified that the increase of PKMDR lifetime with concentration in the solution is an artifact due to the set up of the fluorescence spectrometer, then why not measure the lifetime on a microscope with solution droplets, which is supposedly free of this artifact? Since Figure 1c is an important figure demonstrating the property of the dye, it will improve the paper if one includes a figure free of such artifacts. In addition, a seemingly non-monotonic curve will confuse the downstream interpretation of the data for first time readers.

Response:

We appreciate the reviewer's suggestion.

Cuvette-based fluorometer measurements are regarded as more optically rigorous than microscopy-based measurements due to the higher signal, higher stability, and precise concentrations. That was why we chose it in the very beginning.

Following your suggestions, we tried to measure the lifetime on a microscope with solution droplets. However, we encountered challenges in obtaining reliable lifetime data under these conditions. The primary difficulty arose from the lack of a well-defined focal plane within the droplet for precise lifetime measurements. The dominant signal originated from the reflection at the glass-liquid interface, which precluded accurate lifetime determination at the intended sample plane.

This limitation underscores why fluorescence spectrometer is the gold standard for detecting various parameters (such as fluorescence lifetime, quantum yield, etc.) of fluorescent molecular solutions. Admittedly, artifacts in lifetime measurements arising from radiative energy transport (RET) in fluorescence spectrometer have been previously reported (Kelley & Kelley, 2022). That said, we still think this spectrometer data are the cornerstone of this work.

We have made the following changes to the main text:

To test this hypothesis, we measured the fluorescence intensity and lifetime of **PKMDR** at various concentrations of DMSO using a fluorescence spectrophotometer, whose cuvette-based setup offers accurate measurements. The relationship between fluorescence intensity and dye concentration deviated from linearity starting at 20 μM , indicating self-quenching behavior (Supplementary Fig. S1a)²⁸. Accordingly, the fluorescence lifetime of **PKMDR** sharply declined above 20 μM (Figure 1b), collectively supporting the emergence of concentration-induced quenching. Notably, while **PKMDR** was mainly at a dispersed state in solution below 20 μM , there is a known artifact of radiative energy transport (RET). Within the long light path of the cuvette, the re-absorption of the emitted fluorescence photon at another dye molecule would give rise to an apparent increase of lifetime, accounting for the fluorescence lifetime increase between 1 μM to 20 μM ^{29,30}.

In Figure 1e-h, the data were not fully presented in the original manuscript, so we have made the following changes to the main text:

Comment:

2. The authors wrote in the text that TMRM suffers from non-monotonic intensity-concentration relationship, which I think is too general a statement. TMRM can operate in either quenching and non-quenching mode, whose properties have been well-characterized in the literature. I would rephrase this comment in the text in reference to the literature.

Response:

We thank the reviewer for raising this question. We agree that the two modes of TMRM are not really its drawback. We therefore deleted the related sentence in the main text:

Furthermore, ~~PKMDR lifetime decreases monotonically with concentration, simplifying experimental design and avoiding the misinterpretation risks inherent in TMRM's non-monotonic intensity-concentration relationship.~~

Comment:

3. FLIM of NADH does not measure MMP, hence I would not use it to support the measurement of PKMDR. I would rephrase it in the text.

Response:

We thank the reviewer for pointing this out. We have deleted the related words in the main text:

Conflicting results have been reported between **TMRM** measurements (showing MMP gradient decreasing from the central region to the periphery) and those obtained through **JC-1** imaging ~~and NADH FLIM~~ (demonstrating an inverse peripheral-to-central gradient)⁵⁴⁻⁵⁷. Our analysis aligns closely with the spatial patterns observed in **JC-1** ~~and NADH FLIM~~ studies, which providing support and new insights for the study of spatial heterogeneity of MMP.

Comment:

4. The inclusion of equations are very helpful. For eqn (2), why is the chi-squared computed in the frequency domain, shouldn't it be calculated in the time domain?

Response:

We are grateful to the reviewer for raising this important point. χ^2 is calculated in the time domain. We have corrected the previously inaccurate equation in the main text:

$$\chi^2 = \sum_{k=1}^n \frac{[N(t_k) - N_c(t_k)]^2}{N(t_k)} \quad (2)$$

$N(t_k)$: Measured fluorescence decay function

$N_c(t_k)$: Calculated decay function

n: Evaluated across the number of data points

Kelley, A. M., & Kelley, D. F. (2022). Comment on “Dependence of the Fluorescent Lifetime τ on the Concentration at High Dilution”. *The Journal of Physical Chemistry Letters*, 13(51), 11942-11945. doi:10.1021/acs.jpcllett.2c02677